# Flow Along the $K$-Amplitude for Generative Modeling

**Weitao Du**[1]    **Jiasheng Tang**[1]    **Shuning Chang**[1]
**Yu Rong**[1]    **Fan Wang**[1]    **Shengchao Liu**[2,3]

[1]Alibaba DAMO Academy    [2]University of California, Berkeley
[3]The Chinese University of Hong Kong
duweitao.dwt@alibaba-inc.com, scliu@cuhk.edu.hk

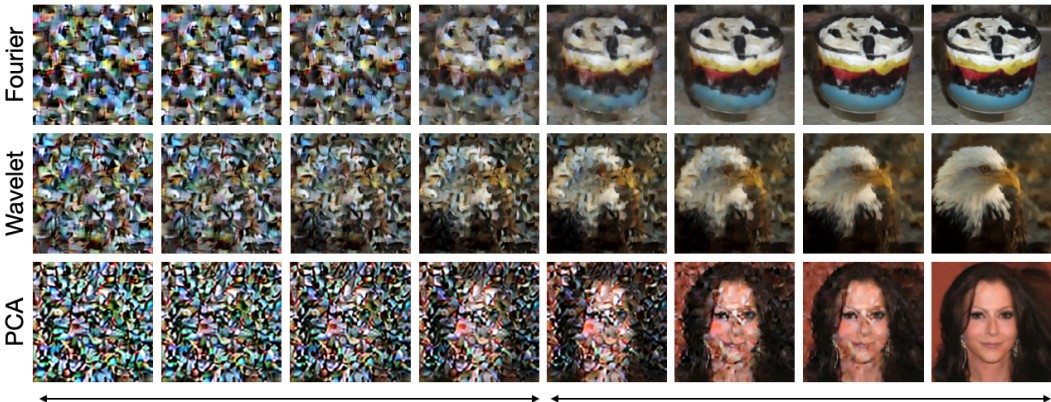

Figure 1: Generation using K-Flow with three $K$-amplitude decompositions: Fourier, Wavelet, and PCA.

## Abstract

In this work, we propose K-Flow, a novel generative learning paradigm that flows along the $K$-amplitude domain, where $k$ is a scaling parameter that organizes projected coefficients (frequency bands), and amplitude refers to the norm of such coefficients. We instantiate K-Flow with three concrete $K$-amplitude transformations: Fourier transformation, Wavelet transformation, and PCA. By incorporating the $K$-amplitude transformations, K-Flow enables flow matching across the scaling parameter as time. We discuss six properties of K-Flow, covering its theoretical foundations, energy and temporal dynamics, and practical applications. Specifically, from the perspective of practical usage, K-Flow allows for steerable generation by controlling the information at different scales. To demonstrate the effectiveness of K-Flow, we conduct experiments on both unconditional and conditional image generation tasks, showing that K-Flow achieves competitive performance. Furthermore, we perform three ablation studies to illustrate how K-Flow leverages the scaling parameter for controlled image generation. Additional results, including scientific applications, are also provided.

## 1 Introduction

Generative Artificial Intelligence (GenAI) represents a pinnacle achievement in the recent wave of AI. This field has evolved from foundational methods such as autoregressive models (AR) (Radford, 2018), energy-based models (Hinton, 2002; Carreira-Perpinan & Hinton, 2005; LeCun et al., 2006; Gutmann & Hyvärinen, 2010; Song & Kingma, 2021), variational auto-encoders (Kingma, 2013), and generative adversarial networks (Goodfellow et al., 2014), diffusion model (Ho et al., 2020; Du et al., 2023), to the most cutting-edge flow-matching (FM) framework (Lipman et al., 2022; Liu et al., 2022b; Albergo & Vanden-Eijnden, 2022).

Among these, flow matching (FM) stands out as a density transport method that converts an initial simple distribution into a complex target distribution through continuous-time flow dynamics. For instance, in the context of image generation, FM learns to map a random Gaussian distribution to the pixel-space distribution of images. This process, termed continuous *flow*, is governed by a localized *k-dependent vector field* (or velocity field) and produces a *time-dependent density path*, which represents the evolution of the probability distribution over time. As a versatile framework, FM can incorporate a diffusion density path, linking it to established methods such as denoising score matching (DSM) (Vincent, 2011; Song & Ermon, 2019) and the denoising diffusion probabilistic model (DDPM) (Ho et al., 2020).

**Motivation.** Natural data exhibits an inherent frequency structure, with most of its energy concentrated in the low-frequency bands (Abry et al., 1995; Van der Schaaf & van Hateren, 1996). Reflecting this property, empirical evidence (Dieleman, 2024) shows that DDPMs tend to denoise from low to high frequencies when transforming white noise (with a uniform frequency spectrum) into meaningful data, allowing earlier recovery of low-frequency components (Biroli et al., 2024). Conventional FMs, however, exhibit different path characteristics (Sun et al., 2025), and their frequency progression is not quantitatively established (Figure S2). In parallel, from the reconstruction perspective, recent research (Kouzelis et al., 2025; Skorokhodov et al., 2025) advocates for explicit frequency consistency constraints as a regularization strategy for auto-encoders. This supports the expectation that introducing frequency-aware path into generation can achieve generation quality on par with or exceeding that of conventional FMs. These observations point to a clear opportunity: developing generative models that offer fine-grained control in the frequency domain can open new frontiers in both generation quality and applicability, *e.g.*, frequency editing and restoration.

**Key Concepts.** To formalize our approach, we first establish a unified framework that integrates key frequency-related concepts from the literature, including Fourier frequency analysis and multi-scale transformations. Central to this framework is the introduction of $K$-amplitude space, parameterized by a scaling parameter $k$. The *scaling parameter $k$* is defined as a systematic measure for organizing frequency bands (or coefficients) across different physical systems and processes (Cardy, 1996; Luijten & Blöte, 1996; Behan et al., 2017; Bighin et al., 2024)[1]. Within this framework, we define *amplitude* as the norm of coefficients obtained by projecting data onto bases corresponding to different scaling parameters $k$, forming what we term the *$K$-amplitude space*, or equivalently, *scaling-amplitude space*.

**Our Method.** Such an understanding of scaling parameter and $K$-amplitude space inspires a new paradigm for generative modeling, which we term **K Flow Matching (K-Flow)**. In essence, K-Flow performs flow along the $K$-amplitude. There are two main components in K-Flow, and the first is the $K$-amplitude decomposition. The $K$-amplitude decomposition encompasses a family of transformations through a linear basis in the $K$-amplitude space, and in this work, we explore three types: Wavelet, Fourier, and principal component analysis (PCA) decomposition, as illustrated in Figure 1. Specifically, K-Flow first applies a $K$-amplitude transformation to project data from the spatial domain into the $K$-amplitude space, where we formulate a novel stochastic interpolant that naturally accommodates the hierarchical structure. In Section A, we provide a comprehensive analysis of K-Flow through six properties, from theoretical foundations (a & b), energy and temporal dynamics (c & d) to practical applications (e & f), with a detailed pipeline illustrated in Figure 2.

**Our Results.** We demonstrate the effectiveness of K-Flow through extensive experiments on generation tasks. Qualitatively, our ablation studies reveal the model's scaling controllability that aligns with our theoretical motivation, enabling two key capabilities: (1) efficient class-conditional generation with minimal guidance, where class information is only required during low-$k$ inference stages; and (2) unsupervised frequency editing through various discretizations of the scaling parameter $k$. Quantitatively, K-Flow achieves state-of-the-art or comparable performance in both unconditional generation and training-free image restoration tasks across natural image and scientific datasets.

---

[1] We distinguish "scaling parameters" in the context of parameterization from "scale" in general discussions

## 2 BACKGROUND

In this section, we first formalize the concept of $K$-amplitude Decomposition, a process governed by a scaling parameter $k$. In the method section, we then detail how our generative framework is designed to operate by flowing along this decomposition, progressively reconstructing the signal.

### 2.1 SCALING PARAMETER $k$, AMPLITUDE, AND $K$-AMPLITUDE DECOMPOSITION

Our data generation framework leverages the implicit hierarchical structure of the data manifold. By 'implicit', we refer to the hierarchical characteristics that emerge when a generalized $K$-amplitude decomposition is applied, transitioning the representation from the original data space to the $K$-amplitude space. Illustrations are in Figure 2. More formally, we represent data as a signal $\phi :$ $\mathbb{R}^d \to \mathbb{R}^m$, or a finite discretization of $\mathbb{R}^d$ and $\mathbb{R}^m$, where this signal function is equivalent to a vector. For example, In the case of image data, each pixel of one RGB channel can be viewed as a signal mapping from the spatial grid $\mathbb{R}^2$ to a pixel intensity value in $\mathbb{R}^1$. Combining the three channels, they form a vector-valued signal from $\mathbb{R}^2$ to $\mathbb{R}^3$. An alternative approach is to consider data as a high-dimensional vector $\mathbb{R}^{d \times m}$. However, treating data as signal functions better fits the decomposition framework in this work.

Without loss of generality, we take dimension $m = 1$ for illustration. A $K$-amplitude decomposition involves the decomposition of a function using a complete basis set $\{e_j\}_{j=1}^n$, where $n$ can be infinite. We introduce a scaling parameter $k$, which partitions the set $\{e_i\}_{i=1}^n$ into subsets: $\{e_i\}_{i=1}^n = \bigcup_k \{e_k\}$, each with $n_k$ basis. Hence, signal $\phi$ is expressed as:

$$\phi = \sum_k \phi_k, \tag{1}$$

where $\phi_k := \sum_{j=1}^{n_k} (\phi \cdot e_{jk}) e_{jk}$ for $e_{jk} \in \{e_k\}$. Inspired by the concept of frequency amplitude, we refer to the norm of $\phi_k$ as the $K$-amplitude. The parameter $k$ is termed the scaling parameter as we expect the natural scaling law exists in well-structured data: the amplitude decays as the value of $k$ increases (Field, 1987). We define $K$-amplitude decomposition (or equivalently, $K$-amplitude transform) $\mathcal{F}$ as the map that sends $\phi$ to the collection of $\phi_k$, and denote the collection of all $\{\sum_{j=1}^{n_k} (\phi \cdot e_{jk}) e_{jk}\}$ as $\mathcal{F}\{\phi\}(k)$. Then,

$$\mathcal{F}\{\phi\} := \bigcup_k \mathcal{F}\{\phi\}(k). \tag{2}$$

We further assume that this transform has an inverse, denoted by $\mathcal{F}^{-1}$.

**Splitting Probability.** Denote the probability of data as $p_{\text{data}}$, then the transformations $\mathcal{F}$ and $\mathcal{F}^{-1}$ induce a probability measure on the associated $K$-amplitude space. In particular, we denote the induced splitting probability of $\phi_k$ as $p(k)$ for each scaling parameter $k$.

In this work, we explore three types of $K$-amplitude decomposition: Wavelet, Fourier, and principal component analysis (PCA). In Section 2.2, we will provide a classic example using the Fourier frequency decomposition on the three-dimensional space. This example serves to illustrate the construction of the scaling parameter $k$ and $K$-amplitude.

### 2.2 EXAMPLE: FOURIER AMPLITUDE DECOMPOSITION

Suppose the data $\phi : \mathbb{R}^3 \to \mathbb{R}$, is drawn from a certain function distribution $p_{\text{data}}$. The challenge of directly fitting the distribution $p_{\text{data}}$ is often complex and computationally demanding. Fourier frequency decomposition, however, offers a powerful technique to address this challenge by transforming $\phi$ into the Fourier space or Fourier domain. In what follows, we will use the terms 'space' and 'domain' interchangeably.

By applying Fourier frequency decomposition, we express $\phi$ as a sum of its frequency components. This transformation can potentially unveil the hidden structure within the distribution $p_{data}$, which is not apparent in the spatial or time domain, and it is thus beneficial for understanding the underlying patterns in the data manifold. To illustrate, the continuous Fourier transform $\mathcal{F}$ of data $\phi(x, y, z) :$ $\mathbb{R}^3 \to \mathbb{R}$ is expressed as:

$$\mathcal{F}\{\phi\}(k_x, k_y, k_z) = \int_{-\infty}^{\infty} \int_{-\infty}^{\infty} \int_{-\infty}^{\infty} \phi(x, y, z) \, e^{-2\pi i(k_x x + k_y y + k_z z)} \, dx \, dy \, dz. \tag{3}$$

After this transformation, the spatial variables $(x, y, z)$ are converted into frequency variables $(k_x, k_y, k_z)$, thereby representing the data in the frequency domain.

Note that the Fourier frequency is characterized by the high-dimensional vector representation $(k_x, k_y, k_z)$. For our purposes, we aim to distill the notion of frequency into a one-dimensional scaling parameter. Namely, we define the scaling parameter $k$ as the diameter of the expanding ball in Fourier space: $k = \sqrt{k_x^2 + k_y^2 + k_z^2}$. This definition of $k$ provides a simple index that captures the overall scaling parameter of the frequency components in all directions. Moreover, we can decompose the Fourier transform $\mathcal{F}\{\phi\}$ into groups indexed by the scaling parameter $k$:

$$\mathcal{F}\{\phi\}(k) = \bigcup_{\sqrt{k_x^2 + k_y^2 + k_z^2} = k} \mathcal{F}\{\phi\}(k_x, k_y, k_z). \tag{4}$$

Intuitively, $\mathcal{F}\{\phi\}(k)$ represents the set of all frequency components that share the same scaling parameter $k$. This grouping allows us to examine the contributions of various spatial frequencies of $\phi$ when viewed through the lens of frequency $k$. Furthermore, $\phi_k$ is just the summation of $\mathcal{F}\{\phi\}(k)$.

On the other hand, we can recover $\phi$ from $\mathcal{F}\{\phi\}$, because the Fourier transform is an invertible operation: $\phi = \mathcal{F}^{-1}\mathcal{F}\{\phi\}$. Such an invertibility establishes the Fourier transform as a valid example of $K$-amplitude decomposition. For discrete data, which inherently possess one highest resolution, the variables $(k_x, k_y, k_z)$ are situated on a discrete lattice rather than spanning the entire continuous space. Consequently, the scaling parameter $k$, derived from these discrete components, is itself discrete and bounded.

## 2.3 FLOW MATCHING

In this work, we primarily focus on the flow matching (FM) generative models and their families (Lipman et al., 2022; Liu et al., 2022b; Albergo & Vanden-Eijnden, 2022). In FM, the flow $\Psi_t$ is defined by solutions of an ordinary differential equation (ODE) system with a time-dependent vector field $\boldsymbol{v}$:

$$\frac{d}{dt}\Psi_t(x) = \boldsymbol{v}_t(\Psi_t(x)), \tag{5}$$

and we focus on the probability transport aspects of $\Psi_t$. In particular, the flow provides a means of interpolating between probability densities within the sample space. Suppose $\Psi_t$ follows an initial probability $p_0$, then for $t > 0$, $\Psi_t$ induces a probability measure $p_t$: $p_t(B) = p_0(\Psi_t^{-1}(B))$, where $B$ is a measurable set. Assume that $\Psi_t$ is differentiable, and define a surrogate velocity at time $t$ as $v_t(x_t, \theta)$ using a deep neural network with parameter $\theta$. Then the vector field matching loss is defined as:

$$\mathcal{L}_{\text{FM}} := \int \int_0^1 dx_0 \, dt \, \left\| \frac{d\Psi_t}{dt}(x_t) - v_t(x_t, \theta) \right\|^2. \tag{6}$$

By aligning the learned vector field with the true gradient field of the frequency decomposition, this loss function ensures robust approximation and reconstruction of the data. Additionally, every interpolation $\pi(x_0, x_1)$ with a time-continuous interpolating function $f_t(x_0, x_1)$ between probabilities $p_0$ and $p_1$ induces a vector field $v_t$ through the continuity equation:

$$\frac{\partial p_t(x_t)}{\partial t} = -\nabla_x \left( p_t(x_t) v_t(x_t) \right), \tag{7}$$

and $v_t$ is explicitly expressed as: $v_t = \frac{1}{p_t} \mathbb{E}_{\pi(x_0, x_1)}\left[\frac{\partial f_t(x_0, x_1)}{\partial t}\right]$. Although explicit matching of $v_t$ via the continuity equation is intractable, flow matching permits a conditional version:

$$\mathcal{L}_{\text{CFM}} = \mathbb{E}_{\pi(x_0, x_1)} \int_0^t dt \, \left\| \frac{\partial f_t(x_0, x_1)}{\partial t} - v_t(x_t, \theta) \right\|^2 + \text{constant}. \tag{8}$$

As detailed in Section 3, our framework reinterprets the time variable $t$ as the scaling parameter $k$. Our goal is to construct a $K$-amplitude-respected $\pi(x_0, x_1)$ with differentiable functions $f_k$.

## 3 METHODOLOGY: K-FLOW

In this section, we introduce K-Flow. It is constructed from the collection of $\mathcal{F}\{\phi\}(k)$, indexed by a specific scaling parameter $k$. As we will demonstrate in Section 3.1, our approach is independent of the specific construction of the invertible transformation $\mathcal{F}$ and the explicit definition of $k$. This flexibility enables us to extend to various $K$-amplitude decompositions.

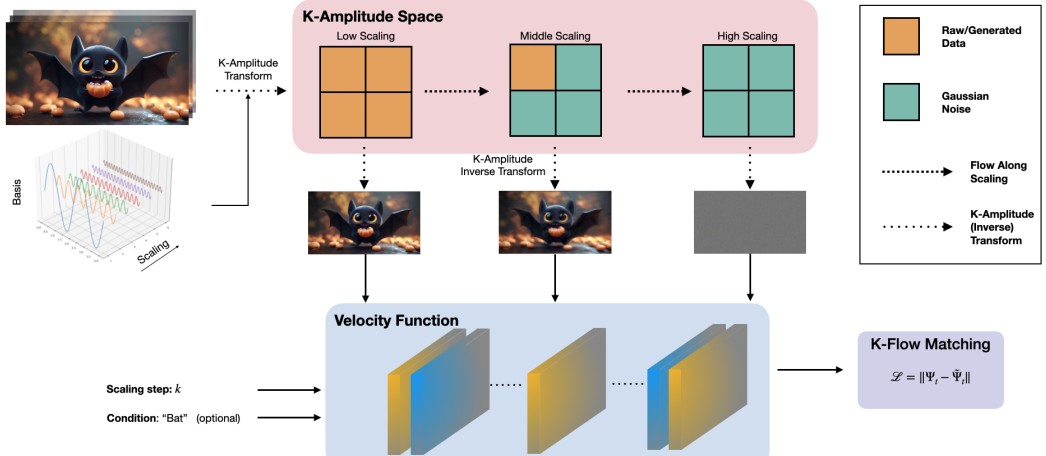

Figure 2: Pipeline of K-Flow. We have a bat figure as the input and three inverted images after three transformations at different granularities.

## 3.1 $K$-AMPLITUDE INTERPOLANTS

According to the concept of stochastic interpolants (Albergo et al., 2023), all flow models can be viewed as constructing stochastic paths that interpolate between a known tractable prior distribution and an unknown target distribution, including flow matching (Lipman et al., 2022), rectified flow (Liu et al., 2022b), and denoising diffusion (Ho et al., 2020). By incorporating the scaling parameter $k$ for $K$-amplitude decompositions, we can formulate a stochastic interpolant that gradually emerges each amplitude component from white noise. Given that $k$ traverses monotonically from zero to a maximum value $k_{\max}$, this process draws a natural analogy to continuous normalizing flows. Since we require $\mathcal{F}$ to be invertible, we can reconstruct the data once the complete spectrum in the $K$-amplitude space is generated.

To build a **continuous flow** $\Psi_k$ out of Equation (1), we explore two paradigms in designing the interpolants: (1) We generalize the original discrete-valued $k$ to continuous values; (2) We ensure that the generation flow, which maps the white noise to the real data, remains invertible such that no information is lost throughout the process. Still taking the three-dimensional signal $\phi(x, y, z)$ and the Fourier transform $\mathcal{F}\{\phi\}$ as an example, we realize the second ingredient by introducing noise padding $\epsilon$ for each $k$ and define the **discrete flow** $\varphi_k$ as follows:

$$\varphi_k = \mathcal{F}^{-1}\left(\mathbb{I}_{\sqrt{k_x^2+k_y^2+k_z^2}\leq k} \cdot \mathcal{F}\{\phi\}(k_x, k_y, k_z) + \left(1 - \mathbb{I}_{\sqrt{k_x^2+k_y^2+k_z^2}\leq k}\right) \cdot \epsilon\right), \quad (9)$$

where $\mathbb{I}$ is the indicator function that selects $K$-amplitude components up to the scaling step $k$. This formulation ensures that for each step $k$, the reconstruction incorporates the relevant K-Flow components of data $\phi$ and pads the rest with noise $\epsilon$. Here, the noise $\epsilon$ is independently drawn from a known distribution (*e.g.*, uniform or Gaussian) for each coordinate $(k_x, k_y, k_z)$. Through this construction, $\phi_k$ serves as a stochastic interpolant for the data $\phi$, ensuring that: $\lim_{k \to k_{\max}} \varphi_k = \phi$, where $k_{\max}$ represents the maximum scaling parameter of data. This limit condition guarantees that as $k$ approaches its maximum value, the reconstructed $\varphi_k$ converges to the original data $\phi$. This behavior is pivotal for the accuracy and fidelity of the generative process. Conversely, $\phi_0$ simply follows the law of a tractable distribution.

**Inter-scaling Interpolant.** Since most of the data we aim to generate is discrete in nature, the $(k_x, k_y, k_z)$ values in the $K$-amplitude decomposition are inherently defined on a lattice. Consequently, the derived scaling parameter $k$ also takes discrete values. This discreteness implies that $\varphi_k$ is originally defined only for discrete values of $k$. However, this discrete flow imposes a limitation: we cannot leverage the powerful flow-matching objective as the optimization framework, which requires taking derivatives with respect to continuous scaling step $k$.

To handle this issue, a straightforward approach is to extend $\varphi_k$ to continuous $k$ by intra-scaling interpolation. That is, we want a continuous flow $\Psi_k$, where $k \in [0, K]$ and satisfy $\Psi_k = \varphi_k$ for integer values of $k$. Let $t := k - \lfloor k \rfloor$ represent the continuous scaling step, where $\lfloor k \rfloor$ denotes the

integer part of $k$. Then, the differentiable interpolation of $\Psi_k$ is:

$$
\Psi_k = \Psi_{\lfloor k \rfloor + t} = \mathcal{F}^{-1} \Bigg( \mathbb{I}_{\sqrt{k_x^2 + k_y^2 + k_z^2} < \lfloor k \rfloor} \cdot \mathcal{F}\{\phi\}(k_x, k_y, k_z) + \mathbb{I}_{\sqrt{k_x^2 + k_y^2 + k_z^2} \geq \lfloor k \rfloor + 1} \cdot \epsilon
$$

$$
+ \mathbb{I}_{\sqrt{k_x^2 + k_y^2 + k_z^2} \in [\lfloor k \rfloor, \lfloor k \rfloor + 1)} \cdot \big( \mu(t) \cdot \mathcal{F}\{\phi\}(k_x, k_y, k_z) + (1 - \mu(t)) \cdot \epsilon \big) \Bigg), \quad (10)
$$

where $\mu(t)$ is a bump function such that $\mu(0) = 1$, $\mu(1) = 0$ and $\mu'(0) = -\mu'(1)$. The antisymmetric property of $\mu'(t)$ ensures that $\Psi_k$ is differentiable from $k$ for all $\mathbb{R}^+$, allowing the flow matching loss and other gradient-based optimization techniques. In Equation (10), we have three components:

1. $\mathbb{I}_{\sqrt{k_x^2 + k_y^2 + k_z^2} < \lfloor k \rfloor}$ applies to the amplitude components up to the integer part of $k$.
2. $\mathbb{I}_{\sqrt{k_x^2 + k_y^2 + k_z^2} \geq \lfloor k \rfloor + 1}$ applies noise padding to components beyond the next integer.
3. $\mathbb{I}_{\sqrt{k_x^2 + k_y^2 + k_z^2} \in [\lfloor k \rfloor, \lfloor k \rfloor + 1)}$ performs linear interpolation of the intermediate amplitude components based on the current $t$.

**Localized Vector Fields.** Instead of directly modeling $\Psi_k$, we pivot our focus to its conditional gradient field, $\frac{d\Psi_k}{dk}$. By concentrating on the gradient field, we facilitate a dynamic view of how $\phi_k$ evolves with respect to $k$. To derive an analytical expression of $\frac{d\Psi_k}{dk}$ conditioned on a given instance pair of data and noise: $(\phi, \epsilon)$, in what follows, we assume that $\mathcal{F}$ is a linear transform. Then, following Equation (10), we have the conditional vector field:

$$
\frac{d\Psi_k}{dk}(\phi, \epsilon) = \mathcal{F}^{-1} \Big( \mathbb{I}_{\sqrt{k_x^2 + k_y^2 + k_z^2} \in [\lfloor k \rfloor, \lfloor k \rfloor + 1)} \cdot \mu'(t)(\epsilon - \mathcal{F}\{\phi\}(k_x, k_y, k_z)) \Big), \quad (11)
$$

for $k \in [\lfloor k \rfloor, \lfloor k \rfloor + 1)$ and $t = k - \lfloor k \rfloor$. Then, following Equation (8), the training objective of K-Flow is to learn the unconditional vector field in Equation (5) by the conditional flow matching:

$$
\mathcal{L}_{\text{K-Flow}} := \mathbf{E}_{\phi_0} \int_0^K d\phi_0 \, dk \, \left\| \frac{d\Psi_k}{dk} - v_k(\Psi_k, \theta) \right\|^2. \quad (12)
$$

By examining Equation (10) closely, we observe that the vector field is naturally localized around a subset of points in the $K$-amplitude space that satisfy $\sqrt{k_x^2 + k_y^2 + k_z^2} \in [\lfloor k \rfloor, \lfloor k \rfloor + 1)$. This localization means that the reconstruction at any given $k$ primarily involves $K$-amplitude components within a narrow frequency band around $k$. Compared with the flow scheme in the pixel space, the $K$-amplitude in K-Flow reduces the optimization complexity by restricting the conditional vector field to be within a sub-manifold for each $k$. This sub-manifold is of low dimensionality, allowing for more focused updates and reducing the optimization space's dimensionality at each step. We will investigate how this localized conditional vector field affects the generation path in Section B. The inference computational complexity of our method is discussed in section D.

We can further generalize the interpolation interval from $(\lfloor k \rfloor, \lfloor k \rfloor + 1)$ to $(k_m, k_n)$, where $k_m$ and $k_n$ are two integers such that $k_m < k_n$. This adjustment broadens the range for intermediate amplitude components from $\sqrt{k_x^2 + k_y^2 + k_z^2} \in [\lfloor k \rfloor, \lfloor k \rfloor + 1)$ to $\sqrt{k_x^2 + k_y^2 + k_z^2} \in [k_m, k_n)$. For example, for our experiments, we partition the $K$-amplitude into two or three parts. See Section D for detailed implementations of these partitioning strategies.

### 3.2 EXAMPLES OF $K$-AMPLITUDE TRANSFORMATION

As we can see from Equation (1), all $K$-amplitude decompositions are achieved through expansion across a complete set of basis functions. However, the behavior of a $K$-amplitude decomposition (transform) can vary significantly depending on the choice of basis functions. Besides the Fourier transform introduced in Section 2, we provide two representative examples of $K$-amplitude decomposition: Wavelet transformation, and PCA transformation. More details are in Section D.

**Wavelet Transform.** Wavelet decomposition (transform) deals with data that are not only scaling-localized but also spatially localized. The scaling parameter of wavelet transform is closely related to the notion of multi-resolution analysis (Mallat, 1989), which provides a systematic way to decompose a signal into approximations and details at successively finer scales. This hierarchical decomposition is achieved through a set of scaling functions $\omega(x)$, and wavelet functions $\psi(x)$, which

together serve as basis functions for the wavelet transformation. More precisely, in the wavelet transform, a signal $f(t)$ is expressed as a sum of scaled and translated versions of these basis functions times the corresponding coefficients $c$ and $d$:

$$f(t) = \sum_j c_{k_0,j}\omega_{k_0,j}(t) + \sum_{k \geq k_0} \sum_j d_{k,j}\psi_{k,j}(t), \tag{13}$$

where $\omega_{k_0,j}(t)$ and $\psi_{k,j}(t)$ are the scaled and translated scaling and wavelet functions, respectively. The index $j$, which originally denotes the translation parameter, groups the basis within each fixed scaling parameter $k$ naturally. Let $\phi_k := \sum_j d_{k,j}\psi_{k,j}$ for $k > k_0$ and $\phi_k := \sum_j c_{k_0,j}\omega_{k_0,j}$ for $k = k_0$, then eq. 13 is just one realization of $K$-amplitude decomposition. Concrete formulas for different families of wavelet bases, such as Daubechies (db), can be found in Section D.

In this article, we employ the discrete version of wavelet transform (DWT) as our $K$-amplitude transformation $\mathcal{F}$, which shares the linearity property with the Fourier transform with a bounded scaling parameter $k$, providing a structured yet flexible means of decomposing discrete data.

**Data-dependent PCA Transform.** Note that Fourier and wavelet decompositions are nonparametric k-amplitude decompositions that are independent of data. While these transformation methods are powerful, we also aim to find data-dependent decompositions that can capture common characteristic features specific to a dataset. This motivation leads to principal component analysis (PCA), a technique widely used for the low-dimensional approximation of the data manifold and vision features (Izenman, 2012; Chen et al., 2024). Please consult Section D for the $K$-amplitude realization of PCA transform.

**$K$-transform without the VAE's latent.** Our current $K$-ampltitude decomposition operates channel-wise on the given input representation (a VAE's latent). We also note that other efficient representations of images (like RGB to YCbCr with sparse discrete cosine transform Nash et al. (2021); Ning et al. (2025)) could yield a superior input space for our K-decomposition. Given that our current results are built upon established pre-trained VAEs, fully integrating a new transform space would necessitate training new auto-encoder models, and we see this as a very promising avenue for future investigation.

## 4 EXPERIMENTS

We conduct a comprehensive experimental evaluation of K-Flow focusing on its technical innovations in $k$-amplitude adapted generation. Using standard backbone architectures, we perform extensive experiments spanning image and molecular assembly generation tasks and $k$ scaling guidance editing and restoration. Complete implementation details, experimental configurations, ablation studies, and scientific generation tasks are provided in Section E (Algorithm 1) and Section D.

### 4.1 IMAGE UNCONDITIONAL AND CONDITIONAL GENERATION

**Image Unconditional Generation** The first task is to generate random samples after fitting a target data distribution, which is typically concentrated around a low-dimensional submanifold within the ambient space.

**Dataset and Metrics.** We conduct experiments on the CelebA-HQ (Karras, 2017) dataset with the resolution of $256 \times 256$. To evaluate the performance of our proposed method, we employ two metrics: the Fréchet Inception Distance (FID) (Heusel et al., 2017), which evaluates the quality by measuring the statistical similarity between generated and real images, and Recall (Kynkäänniemi et al., 2019), which measures the diversity of the generated images.

Table 1: Unconditional generation on CelebA-HQ.

| Model | FID↓ | Recall↑ |
|---|---|---|
| K-Flow, Fourier-DiT L/2 (Ours) | 5.11 | 0.47 |
| K-Flow, Wave-DiT L/2 (Ours) | **4.99** | 0.46 |
| K-Flow, PCA-DiT L/2 (Ours) | 5.19 | 0.48 |
| LFM, ADM (Dao et al., 2023) | 5.82 | 0.42 |
| LFM, DiT L/2 (Dao et al., 2023) | 5.28 | 0.48 |
| WaveDiff, DiT L/2 (Phung et al., 2023) | 5.38 | 0.44 |
| FM (Lipman et al., 2022) | 7.34 | - |
| LDM (Rombach et al., 2022) | 5.11 | **0.49** |
| LSGM (Vahdat et al., 2021) | 7.22 | - |
| WaveDiff (Phung et al., 2023) | 5.94 | 0.37 |
| DDGAN (Xiao et al., 2021) | 7.64 | 0.36 |
| Score SDE (Song et al., 2020) | 7.23 | - |

**Results.** Table 1 summarizes the comparison between our proposed K-Flow model and other generative models. For a fair comparison, both the baseline ordinary flow matching (LFM (Dao et al.,

2023)) and our K-Flow flow utilize the same VAE's latent from (Rombach et al., 2022). Our K-Flow employs a lightweight MoE variant of the Diffusion Transformer (DiT L/2 (Peebles & Xie, 2023)) as the backbone model, while LFM uses the standard DiT L/2 architecture. We can observe that (1) K-Flow achieves the best performance in FID, especially w/ the db6-based wavelet K-Flow. (2) Although the latent diffusion model (Rombach et al., 2022) gets the highest score in Recall (diversity), the Fourier and PCA-based K-Flow is comparable with the ordinary latent flow matching.

**Image Class-conditional Generation** Then we explore how $K$-amplitude decomposition behaves when the generation path is conditioned on class labels, where the class label (*e.g.*, dog, cat, fish, etc) delegates the low-scaling information of each image. This investigation could potentially pave the way for multi-scaling control, where different scaling components are influenced by specific caption information. We list the detailed class-conditional generation algorithm in Section E.

Table 2: Class-conditional generation on ImageNet.

| Model | FID↓ | CDR↓ | Recall↑ |
|---|---|---|---|
| K-Flow, Wave-DiT L/2 (Ours) | 17.8 | - | 0.56 |
| + cfg=1.5 | 4.49 | - | 0.44 |
| K-Flow, PCA-DiT L/2 , cfg=1.5 | 4.19 | - | 0.43 |
| K-Flow, Fourier-DiT L/2 (Ours) | 13.4 | - | 0.57 |
| + cfg=1.5 | **2.73** | **1.49** | 0.45 |
| LFM, DiT L/2 | 14.0 | - | 0.56 |
| + cfg=1.5 | 2.85 | 3.25 | 0.42 |
| LDM-8 (Rombach et al., 2022) | 15.51 | - | **0.63** |
| LDM-8-G | 7.76 | - | 0.35 |
| DiT-B/2 Peebles & Xie (2023) | 43.47 | - | - |
| VAR-d16 Tian et al. (2024) (cfg=2.0) | 3.30 | - | 0.51 |
| FlowAR-B Ren et al. (2024) (cfg=2.0) | 3.17 | - | 0.52 |
| FlowAR-L Ren et al. (2024) (cfg=2.4) | **1.9** | - | 0.57 |

**Dataset and Metric.** We use ImageNet with resolution $256 \times 256$ as the middle-size conditional generation dataset (Deng et al., 2009). Beyond evaluating the unconditional FID for the ImageNet dataset, we are also interested in studying how the class control interacts with scaling generation in a quantitative manner.

**Results.** The results are presented in Table 2. Our primary focus for the FID metric is the classifier-free guidance inference method applied to flow matching models. The data indicates that K-Flow achieves results comparable to LFM. In terms of the recall metric, which assesses the diversity of the generated distribution, our model outperforms the standard LFM. This improvement may be attributed to the fact that the inference path of K-Flow includes a greater number of dimensions during the low-scaling period, as discussed in Section B.1. We extend our comparison to include two modern multi-scale auto-regressive models, VAR (Tian et al., 2024) and FlowAR (Ren et al., 2024), with architectural and performance details provided in Table S2. Among models of a comparable scale, our $K$-**Flow (Fourier)** achieves a superior FID of **2.72**, demonstrating the efficacy of our framework. The state-of-the-art FlowAR-L (FID 1.90) serves as an upper-bound reference, as it utilizes a significantly larger model and a different VAE, placing it in a different comparison category.

## 4.2 Image Controllable Class-conditional Generation

The latent flow matching model can implicitly learn low- and high-resolution features (Dao et al., 2023), but the boundary between each resolution is vague, and we cannot explicitly determine which timestep in the inference process corresponds to a specific resolution or frequency. In comparison, our proposed K-Flow enables finer-grained controllable generation. As demonstrated in Figures 3a and 4, K-Flow effectively preserves high-frequency details even when class conditions are omitted during the last 70% of scaling steps, whereas ordinary latent flow exhibits significant blurring. To quantitatively validate this observation, we analyze the conditional discrimination ratio (CDR, formally defined in Section F). From Table 2, our model maintains a CDR close to one, indicating robust performance regardless of high-scale condition omission, while conventional LFM shows significantly higher CDR, suggesting performance degradation. The class-dropping experiment provides quantitative evidence that K-Flow learns a disentangled generative path by encoding high-level semantics into low K-amplitude bands, highlighting its potential for improved computational efficiency by allowing control conditions to be omitted during later synthesis stages.

## 4.3 Image Scaling-controllable Generation and Restoration

Our method guarantees that the generation path is disentangled with respect to $k$ (check Section B). This allows us to control initial noise at each scaling level (see Algorithm 2), enabling unsuper-

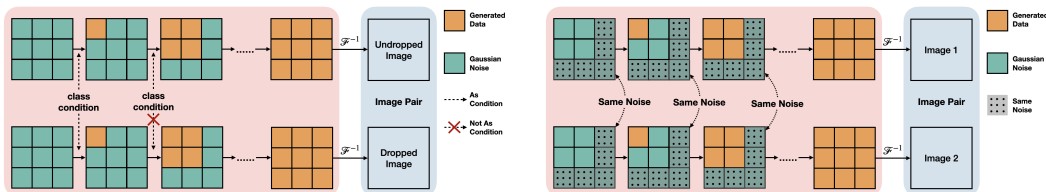

(a) Controllable class-conditional generation.   (b) Scaling-controllable generation (low scaling).

Figure 3: Pipeline of two ablation studies on controllable generation.

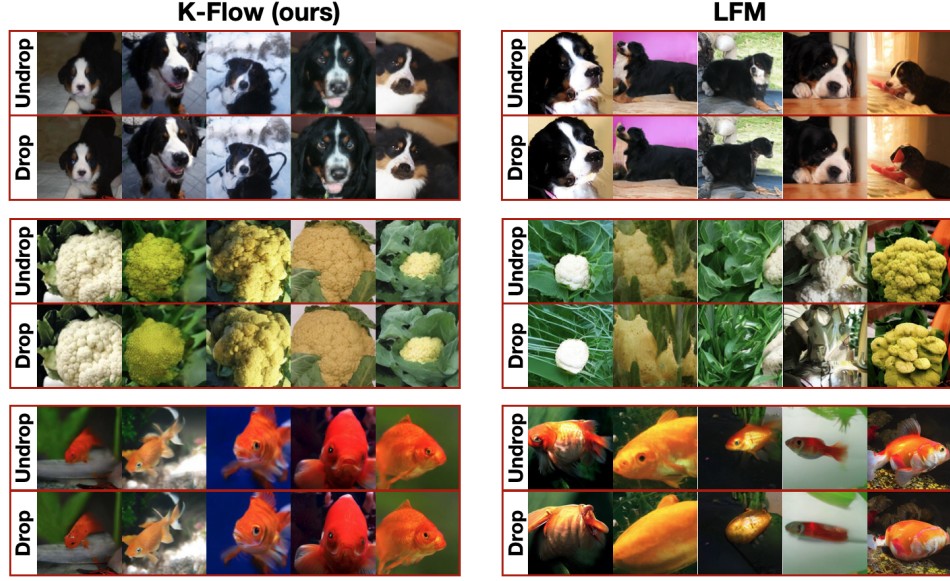

Figure 4: Results of controllable class-conditional generation. 'Drop' means we drop the class conditions during the last 70% scaling steps, while 'undrop' means we keep the condition all the time.

vised editing of different scaling components. We also conduct ablation studies on **Preserving Low Scaling, Modifying High Scaling**. Please check Section F for more details and visualizations.

**Preserving High Scaling, modifying Low Scaling.** This scaling-controllable generation pipeline is illustrated in Figure 3b. It involves sampling multiple images while ensuring that the noise in the high-scaling components remains consistent across all samples. In scaling-controllable image generation, the goal is to maintain consistency in the high-scaling details while allowing variations in the low-scaling context among the generated images, thus this allows K-Flow to achieve unsupervised steerability in a finetuning-free manner.

The results on CelebA are presented in Figure 5, where we apply a pretrained Daubechies wavelet (db6-based) K-Flow. It can be observed that facial details, such as eyes, smiles, noses, and eyebrows, remain consistent within each group of images. In contrast, the low-scaling components, including background, gender, age, and hairstyle, vary across the images within the same group. These qualitative results demonstrate how frequency bands naturally correspond to semantic features - facial details persist in high-frequency components while attributes like global background and overall appearance vary in low-frequency components. When applying the same editing protocol to conventional LFM, the results show no such organized frequency-semantic correspondence (Figure S10), highlighting the advantage of K-Flow.

**Image Restoration.** While traditional semantic metrics cannot directly assess unsupervised frequency-based editing, we quantitatively validate our scaling-aware generation path via image restoration tasks, where frequency-specific changes are objectively measurable. This includes super-resolution and deblurring. From Table S6 in Appendix, K-Flow achieves state-of-the-art performance in terms of PSNR and SSIM metrics on the CelebA dataset. This demonstrates that our frequency-domain formulation enables accurate high-frequency reconstruction. Detailed experimental settings and algorithms are provided in Section F.

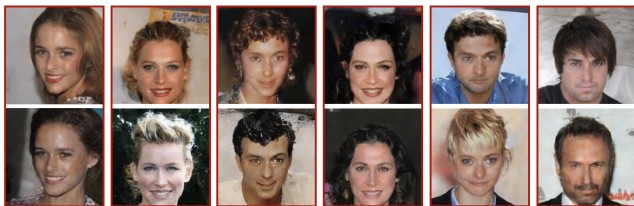

Figure 5: Results of scaling-controllable generation: Each column presents an image pair that shares high-frequency components while exhibiting distinct low-frequency characteristics.

## 5 CONCLUSION

In this paper, we introduce K-Flow Matching (K-Flow), an efficient flow-matching model that flows along the $K$-amplitude for generative modeling. K-Flow naturally generalizes the multi-scales of data (*e.g.*, multi-resolution or frequencies in images) to multi-scales in the $K$-amplitude space.

**Limitations and Future Directions.** As we have verified the effectiveness of K-Flow exclusively on image generation tasks, moving forward, three promising directions are worth exploring. (1) Multi-modal generation that includes tasks such as large-scale data generation guided by dense captions. This is an extension of our class-dropping experiment, which could better showcase the efficient steerability of K-Flow by aligning images with natural language inputs. (2) We outline six properties of K-Flow in Section 1, *e.g.*, the amplitude naturally corresponds to energy. While Section 3 briefly discusses how energy is represented in K-Flow, this aspect has not been explored in depth. We believe that such energy term holds potential for integration with the utility of energy-based models in future work. (3) Unifying understanding and generation: Our method demonstrates that generation can be performed in a scale-aware manner, particularly by following a coarse-to-fine frequency ordering (i.e., from low to high frequencies). From a representation learning perspective, low-frequency components naturally encode global semantic features that are closely aligned with those learned in visual understanding tasks. This observation suggests that K-Flow holds potential for bridging the gap between visual understanding and generation, by incorporating representations pre-trained on understanding tasks into the generative process. We regard this as a compelling direction for future exploration toward unified visual modeling.

## ETHICS STATEMENT

This research adheres fully to the ICLR Code of Ethics. It does not involve human subjects, personal information, or sensitive data. All datasets and code used or released comply with their respective licenses and terms of use. The contributions of this work are methodological and foundational, raising no concerns related to fairness, privacy, security, or potential misuse.

## ACKNOWLEDGEMENTS

We thank the reviewers for their valuable feedback and suggestions that helped improve this work.

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

## THE USE OF LARGE LANGUAGE MODELS

We employ an LLM to refine the language and edit the draft of this paper, including:

- Correcting grammatical errors, punctuation, and spelling.
- Improving sentence structure to enhance clarity, flow, and readability.
- Suggesting alternative phrasing for more precise and professional academic expression.

All modifications suggested by the LLM were critically reviewed, vetted, and approved by the authors. The final text accurately reflects our own ideas, arguments, and research findings.

## A    PROPERTIES OF K-FLOW

(a) K-Flow provides **a first-principle way to organize the scaling** $k$. Unlike perception-based computer vision tasks, which often favor certain scaling (frequency) bands, a $K$-amplitude based generative model strives for an optimal organization of all scalings to ensure that the final generated sample is of high fidelity. By constructing $K$-amplitude scaling-based vector fields, the integrated flow naturally incorporates all scaling information, and the conditional flow matching training objective provides a perfect trade-off of accuracy-efficiency inside localized scalings. We will also demonstrate how different discretizations of K-Flow with related works, highlighting the connections and integrations with existing methods in the field.

(b) K-Flow enables **multi-scale modeling in the $K$-amplitude space**. Compared to the original data space, such as the pixel space in images, the $K$-amplitude space provides a more natural perspective for defining and analyzing multi-scale information, namely, $K$-amplitude decomposition empowers K-Flow for effective multi-scale modeling. By decomposing the feature representation into multiple scaling components in the $K$-amplitude space, K-Flow associates each scaling with an amplitude. Higher values of $K$-amplitude correspond to higher-frequency information, capturing fine-grained details, while lower values encode lower-frequency information, representing more coarse-grained features. Let us take the image for illustration. Images inherently exhibit a hierarchical structure, with information distributed across various resolution levels. Low-resolution components capture global shapes and background information, while high-resolution components encode fine details like textures, often sparse and localized. By projecting these components into the $K$-amplitude space, K-Flow captures such hierarchical information effectively and naturally, enabling precise modeling of the interplay between scales.

(c) K-Flow supports **a well-defined scale along with energy**. The amplitude is also used to reflect the *energy* level at each scale of the data. In physics, it is proportional to the square of the amplitude. In comparison, for the modeling on the original data space, though we can inject application-specific inductive bias, such as multiple pixel resolutions for images, they do not possess a natural energy concept.

(d) K-Flow interprets **scaling as time**. From elucidating the design space of the traditional flow matching perspective, K-Flow re-defines the artificial time variable (or the signal-to-noise ratio variable proposed in (Kingma et al., 2021)) as the ordering index of frequency space. In this context, the artificial time variable effectively controls the traversal through different levels of a general notion of frequency decompositions, scaling each frequency component appropriately. This perspective aligns with the concept of renormalization in physical systems, where behavior across scales is systematically related.

(e) K-Flow supports the **fusion of intra-scaling and inter-scaling modeling.** K-Flow flows across scaling as time, and namely, K-Flow naturally merges the intra- and inter-scaling during the flow process. Thus the key module turns to the smooth interpolant, as will be introduced in Section 3. This is in comparison with existing works on multi-modal modeling (Burt & Adelson, 1987; Tian et al., 2024; Atzmon et al., 2024), where the special design of the intra-scaling and inter-scaling is required.

(f) K-Flow supports **explicit steerability**. The flow process across scales enables K-Flow to control the information learned at various hierarchical levels. This, in turn, allows finer-grained control of the generative modeling, facilitating more precise and customizable outputs. By understanding and leveraging K-Flow's steerability, its utility can be significantly enhanced across diverse domains,

including Artificial Intelligence-Generated Content (AIGC), AI-driven scientific discovery, and the safe, responsible development of AI technologies.

# B    DISCUSSION

## B.1    FROM CONDITIONAL TO UNCONDITIONAL PATH IN K-FLOW

In Section 3, our frequency-localized path is defined at the conditional level ($\frac{d\Psi_k}{dk}(\phi, \epsilon)$ ) , and it is only related to the unconditional vector field ($v_k(\Psi_k, \theta)$ in eq. (12)) through the equivalence of conditional flow matching and unconditional flow matching at the loss level (Lipman et al., 2022). In this section, we try to study the splitting property of the unconditional $K$-amplitude vector field.

By the $K$-amplitude decomposition, the transformed data probability $p_{data}$ satisfies the telescoping property:

$$p_{data} = p(k_0)p(k_1|k_0)\dots p(k_{max}|k_{max} - 1, \dots, k_0), \tag{14}$$

with $k_0$ and $k_{max}$ denoting the lowest and highest scaling. Then, according to the definition of our proposed K-Flow $\Psi_k$, the interpolated probability at scaling step $t$ is also localized:

$$p_t(\cdot) = p(k_0)\cdots p_t(\cdot|\lfloor k \rfloor, \dots, k_0)p_\epsilon(\lfloor k \rfloor + 1)\cdots p_\epsilon(k_{max}|k_{max} - 1, \dots, k_0), \tag{15}$$

where $p_\epsilon$ denotes the distribution of the initial noise and $t \in [\lfloor k \rfloor, \lfloor k \rfloor + 1)$. Combining Equation (15), the localization property of the bump function, and Lemma 1 of (Zheng et al., 2023), the unconditional vector field has an explicit form: $v_t(\Psi_k) = a_t \cdot \Psi_k + b_t \nabla \log p_t(\Psi_k)$, where $a_t$ and $b_t$ are hyper-parameters determined by the bump function we choose.

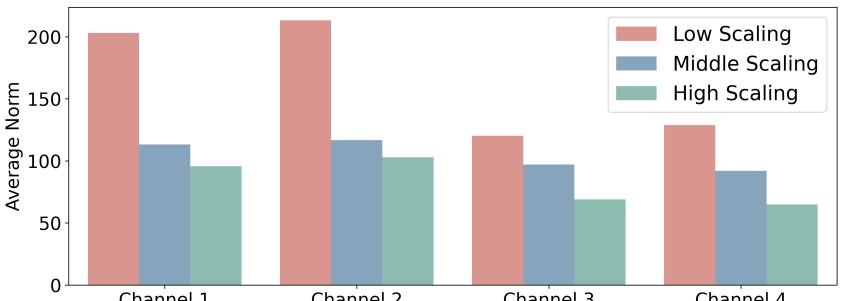

Supplementary Figure S1: On the low-scaling hypothesis. The graph illustrates the relative norm distribution for each scaling component as defined by the wavelet decomposition in the latent space. It can be observed that the low-scaling component exhibits a significantly higher norm (energy), nearly twice that of the high-scaling component.

**Noise Splitting.** A key characteristic of flow models is their deterministic nature after the initial noise sampling. Specifically, once the initial noise is sampled, the flow follows a fixed path to generate the final data sample. According to Equation (15), during scaling step t: (1) the scaling components below $\lfloor k \rfloor$ remain unchanged; (2) the scaling components above $\lfloor k \rfloor$ remain unchanged; (3) The distribution of higher scaling components maintains the same characteristics as their initial noise distribution.

By these observations, we now investigate how segmented initial noise in the K-Flow space influences the final output of the K-Flow flow. Suppose we discretize scaling parameter $k$ into two parts: $\mathcal{F}\{\Psi_k\} = \{\phi_{\text{low}}(k), \phi_{\text{high}}(k)\}$. When flowing along the low-scaling component, the vector field $v_k$ can be re-expressed in a conditional form:

$$v_k(\Psi_k) = v_k(\phi_{\text{low}}(k), c) \tag{16}$$

where constant $c$ represents the (static) initial noise for the high-scaling part. This noise-conditioned property in the k-amplitude domain leads us to explore whether fixing the high-scaling noise and altering the low-scaling noise allows for unsupervised editing of relative low-scaling semantics in an image. Indeed, we observed this phenomenon, the qualitative results will be discussed in Section 4.3.

From Figure 5, we observe that a targeted common high-scaling initial noise guides our K-Flow flow toward generating human faces with similar detail but varying low-level content. See the experiment section for a more detailed analysis.

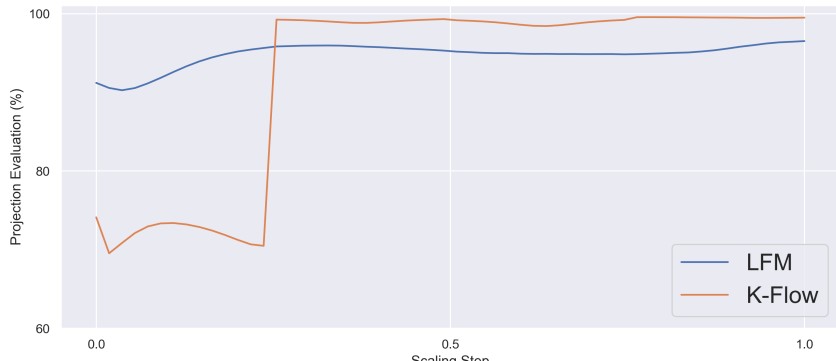

Supplementary Figure S2: **Projection Error Comparison with Different flow Models.** The graph illustrates the PCA projection errors of two trained models throughout the entire inference process, with distinct segments marked by dashed lines. The red and blue lines represent the original latent flow matching (LFM) and the K-Flow with two amplitude components, respectively. The projection error is quantified by the reconstruction error for each generation step from the PCA compression, using the first two principal components. Owing to the scaling-aware nature of our flow, the low-amplitude portion (the initial part of the curve) resides in a relatively high-dimensional space, resulting in higher projection errors for the two-dimensional PCA projection.

### B.2 THE EFFECT OF SCALING STEP $k$ FOR IMAGE RECONSTRUCTION

K-Flow's ability to leverage the low-dimensional structure of data is primarily enabled by its K-Flow localization property. This enables a strategic path through low-dimensional spaces, which can be directly compared with the generation path of conventional flow models. In our model, this path incorporates an explicit frequency hierarchy, which hypothesizes that the low-frequency components - concentrated in the earlier stages of the model - may share more dimensions in common, particularly from a semantic perspective, than the high-frequency components positioned later in the generative process. Conversely, an ordinary flow model may exhibit a more uniform distribution of dimensionality across the entire generative path.

Motivated by this hypothesis, we conduct a case study using PCA to approximate the dimension of the generation trajectory $\{\Psi_k\}_{k=k_0}^{k_{max}}$. As illustrated in Figure S2, we measure how closely the dimension of the generation path aligns with a two-dimensional subspace spanned by the first two components of the model's PCA decomposition, denoted by $\{\tilde{\Psi}_k\}_{k=k_0}^{k_{max}}$. Inspired by (Zhou et al., 2024), the reconstruction ratio is defined by $1 - \|\Psi_k - \tilde{\Psi}_k\|_2 / \|\Psi_k\|_2$. In other words, a higher value of the reconstruction ratio indicates that the model's dimension is closer to two. Therefore, the trend of the error curve with respect to the scaling parameter $k$ reveals a distinct separation in the effective dimension between low- and high-scaling components. Evidently, the low-scaling segments display more semantic consistency and thus, occupy a larger dimensional space, whereas the high-scaling segments converge to a more confined or lower-dimensional structure.

It is important to note that this exploration into the dimensionality of generative paths is practically meaningful. Previous study (Zhou et al., 2024) has shown that the effectiveness of distilling a generative model with fewer steps from a pre-trained diffusion model theoretically depends on the model's dimensionality at each step, as informed by the high-dimensional Mean Value Theorem. The observations from Figure S2 provide empirical support for this concept. Specifically, the ability of K-Flow to maintain a lower-dimensional structure in high-scaling components suggests a promising approach for fast sampling distillation methods.

### B.3 RELATED WORK DISCUSSION

The field of generative modeling has seen significant advancements in recent years, driven by a variety of frameworks, including adversarial generative networks (GAN) (Goodfellow et al., 2014), variable autoencoders (VAE) (Kingma, 2013), and normalizing flows (Papamakarios et al., 2021). In this work, we focus on continuous normalizing flow generative models (Chen et al., 2018), with

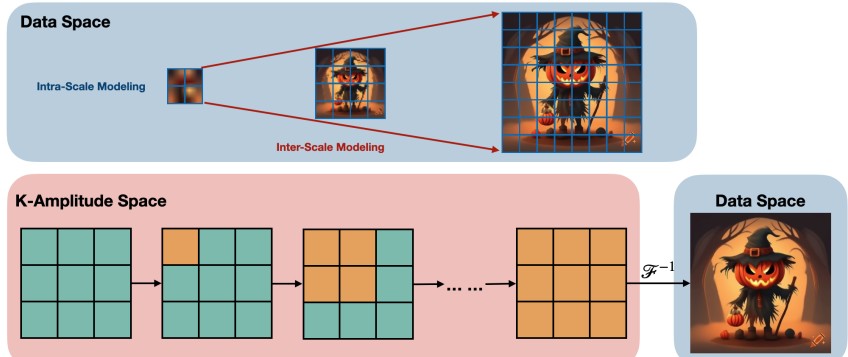

Supplementary Figure S3: Comparison of multi-scale modeling: pixel data space and K-Amplitude space.

particular emphasis on the conditional flow matching training scheme, which originates from the denoising score matching training framework (Vincent, 2011).

Both diffusion models and continuous flow matching models aim to lower the complexity of directly optimizing the log-likelihood of data by introducing an additional stochastic path. However, as proved in (Lavenant & Santambrogio, 2022), the canonical path for diffusion models and rectified flows is not optimal. This realization motivates our introduction of frequency decomposition as a key design element in generative models.

By breaking down the formula of our K-Flow vector field with respect to the scaling parameter $k$, we can summarize three successful factors as general principles for scaling modeling.

- A good $K$-amplitude decomposition can leverage the problem's inherent biases towards certain scaling bands. For generative tasks, it is crucial that all K-Flow bands are effectively modeled to ensure the generation of high-quality, controllable outputs. In addition, the computational resources required may vary between different scales, thus necessitating careful consideration of resource allocation.
- Modeling within each scaling component, which is formulated in our K-Flow-localized vector fields.
- Modeling bridges along different scalings, which is achieved through our flow ODE and the (time) K-Flow embedding block for the U-Net or DIT architecture.

This approach to inter- and intra-modeling for $K$-amplitude is also applicable to scenarios emphasizing certain frequencies or scalings. For instance, (Li et al., 2024a) enhanced oscillatory motion control in video generation by discarding the high-frequency component of the Fourier decomposition. As discussed in Section 3, the scaling parameter of spatially localized wavelet (multi-resolution) decomposition is closely linked to image resolution. Notable contributions in this domain include (Atzmon et al., 2024) and (Lei et al., 2023), which introduced a multi-stage resolution for fine-grained editing, and (Jin et al., 2024), which concentrated on efficient video generation. We provide a systematic review of frequency- or scaling-based generative approaches in Section C, highlighting key developments in this direction.

In related research on auto-regressive modeling, (Mattar et al., 2024) presented wavelets as an effective auto-regressive unit, while (Tian et al., 2024) focused on the scale as a key element for image auto-regression. A significant example is (Phung et al., 2023), which transitioned the latent space from pixel to wavelet space for generative models using wavelet diffusion. However, their method employed the same conditional noising schedule for score matching as traditional diffusion models. In contrast to their approach, our proposed K-Flow integrates wavelet decomposition as a multi-channel module within the neural network architecture for training diffusion models. Additionally, our work extends the notion of wavelet space to the more general $K$-amplitude space.

We also want to highlight another research line that has recently caught the attention: the auto-regressive modeling over the pixel space for image generation. One classic work is VAR Tian et al. (2024). It introduces a hierarchical down and up sample paradigm that models images in a coarse-to-fine manner across multiple resolutions and models the data distribution in an auto-regressive manner. In contrast, our proposed K-Flow integrates the flow paradigm for density estimation and

leverages the $K$-amplitude space as a stronger inductive bias, as illustrated in Figure S3. Another related work is the auto-regressive flow model proposed in (Ren et al., 2024), that implements conditional flow matching sequentially across scales. Although (Ren et al., 2024) shares some common terminology with our work (*e.g.*, scales, flow matching), K-Flow provides significant advantages through its unified flow process, almost architecture-agnostic design, and theoretically grounded frequency domain framework. On the practical implementation side, Unlike (Ren et al., 2024), which requires separate flow matching for each scale and relies on specific architectures (autoregressive transformers that treat scaling as conditional input), our approach implements a single coherent flow that connects all frequency scales during inference while maintaining architecture flexibility.

**Summary.** In summary, K-Flow is a more general framework, with its three key factors potentially benefiting generation-related tasks like super-resolution and multi-resolution editing. For example, (Liu et al.) utilized a learnable Fourier transform to construct a harmonic module in the bottleneck layer of an autoencoder. We provide a comprehensive list of related works in Section C.

### B.4 CONNECTING K-FLOW WITH SSL REPRESENTATION AND GENERATION

From the above discussion, we have seen how pretrained vision models leverage the sparsity and locality of natural data in various $K$-amplitude domains for perception and generation-based tasks. In the realm of unsupervised learning, (Liu et al., 2022a; 2024a; Chen et al., 2024) explore whether generative-based representations, particularly those derived from denoising diffusion models, can achieve parity with contrastive-based representation learning methods for downstream tasks. A key observation from their findings Chen et al. (2024), which aligns with our approach of employing $K$-amplitude decomposition (the PCA instance), is the revelation that the most powerful representations are obtained through denoising within a latent space, such as the compressed PCA space. Another merit of PCA is that denoising along the PCA directions can achieve faster convergence for denoising, which is revealed in (Du et al., 2023).

To transition from unsupervised representation learning to real data generation, incorporating all $K$-amplitude scalings is essential. Rather than compressing or amplifying specific scaling bandwidths, generative tasks require novel organization or ordering of all frequencies. Besides our flow-based frequency generation approach, (Tian et al., 2024) connects different scales (which can be interpreted as the wavelet $K$-amplitudes) using residual connections with an auto-regressive training objective. Residual connections, as a discretization of ordinary differential equations (ODEs) proposed in (Ee, 2017), suggest that (Tian et al., 2024)'s approach can be seen as a special discretization of our K-Flow with a flexible flow matching training objective.

Supplementary Table S1: Comparison among PCA, contrastive, and generative SSL.

|  | Basis Learning | Reconstruction Learning |
|---|---|---|
| PCA SSL | Non-parameterized, Determined By Data | Parameterized |
| Contrastive SSL | Parameterized | N/A |
| Generative SSL | Parameterized | Parameterized |

## C    RELATED WORK ON FREQUENCY, AND MULTI-SCALING

There have been multiple research lines on studying generative modeling, especially in terms of multi-scale modeling. In this work, we would like to summarize them as the following three venues.

### C.1    MULTI-SCALE IN PIXEL RESOLUTION, FLOW AND DIFFUSION

**Laplacian Pyramid and Laplacian Operator.** In mathematics, the Laplacian operator computes the second derivative of a function, emphasizing regions with significant intensity changes, such as edges or high-frequency details. Similarly, the Laplacian Pyramid (Burt & Adelson, 1987) decomposes an image into multiple scales, extracting the low-frequency components (smooth regions) through downsampling. The high-frequency details, such as edges and textures, are modeled as the residuals between adjacent resolution layers. The primary objective of the Laplacian Pyramid is to represent these residuals across scales in a hierarchical fashion.

**LAPGAN (Laplacian Generative Adversarial Networks)** (Denton et al., 2015) adopts the Laplacian pyramid idea into the generative adversarial network (GAN) framework (Goodfellow et al., 2014). By focusing on learning residuals between successive levels of resolution, it effectively generates high-quality super-resolution images.

**SR3 (Super-Resolution via Repeated Refinement)** (Saharia et al., 2022) leverages DDPM (Denoising Diffusion Probabilistic Models) (Ho et al., 2020) and DSM (Denoising Score Matching) (Vincent, 2011; Song & Ermon, 2019) for high-resolution image generation. Specifically, SR3 enhances low-resolution images to high-resolution by utilizing multiple cascaded conditional diffusion models. In this framework, the low-resolution images serve as conditions, and the model's aim is to predict the corresponding high-resolution images as outputs.

**PDDPM (Pyramidal Denoising Diffusion Probabilistic Models)** (Ryu & Ye, 2022) is a follow-up work of SR3, and it improves the model by only modeling one score network. The key attribute to enable this is by adding the fractional position of each pixel to the score network, and such fractional position information can be naturally generalized to different resolutions.

**f-DM** (Gu et al., 2022) is developed concurrently with PDDPM and shares the approach of utilizing only one diffusion model. It distinguishes itself by explicitly applying a sequence of transformations to the data and emphasizing a resolution-agnostic signal-to-noise ratio within its diffusion model design.

**Edify Image** (Atzmon et al., 2024) is a state-of-the-art model capable of generating photorealistic, high-resolution images from textual prompts (Atzmon et al., 2024). It operates as a cascaded pixel-space diffusion model. To enhance its functionality, Edify Image employs a downsampling process that extracts low-frequency components and creates three distinct resolution levels, ranging from low to high frequency, with the original image representing the highest frequency level. Another key innovation of Edify Image is its meticulously crafted training and sampling strategies at different resolutions, utilizing attenuated noise schedules.

### C.2    MULTI-SCALE IN PIXEL RESOLUTION, VAE AND AR

**VQ-VAE2 (Vector Quantized VAE 2)** (Razavi et al., 2019) enforces a two-layer hierarchical structure, where the top layer captures global features such as object shapes and geometry, while the bottom layer focuses on local details like texture. It models data density within the variational autoencoder (VAE) framework(Kingma, 2013) and incorporates an autoregressive (AR) module to enhance the prior for improved generative performance.

**RQ-VAE (Residual-Quantized VAE)** (Lee et al., 2022) integrates recursive quantization into the VAE framework. It constructs a representation by aggregating information across $D$ layers, where the first layer provides a code embedding closely aligned with the encoded representation, and each subsequent layer refines this by reducing the quantization error from the previous layer. By stacking $D$ layers, the accumulated quantization error is minimized, enabling RQ-VAE to offer a coarse-to-fine-grained approach to modeling. For modeling, the general pipeline follows the VAE framework, while each latent code is decomposed into $D$ layers and is predicted in an autoregressive manner.

**VAR (Visual AutoRegressive)** (Tian et al., 2024) introduces a novel paradigm for density estimation by decomposing images into multiple resolutions across various scales. This approach is inspired by the hierarchical nature of human perception, where images are interpreted progressively from global structures to finer details. Leveraging this concept, VAR models the entire image in a coarse-to-fine manner, adhering to the principles of multi-scale hierarchical representation.

### C.3 Multi-Scale in Frequency, AR, VAE, and Diffusion

**WaveDiff (Wavelet Diffusion)** (Phung et al., 2023) leverages the discrete wavelet transform to shift the entire diffusion process into the wavelet spectrum. Its primary objective is to reduce model complexity by operating in the transformed spectrum space instead of the pixel domain.

**PiToMe (Protect Informative Tokens before Merging)** (Tran et al., 2024) is a token merging method designed to balance efficiency and information retention. PiToMe identifies large clusters of similar tokens as high-energy regions, making them suitable candidates for merging, while smaller, more unique, and isolated clusters are treated as low-energy and preserved. By interpreting attention over sequences as a fully connected graph of tokens, PiToMe leverages spectral graph theory to demonstrate its ability to preserve critical information.

**WF-VAE (Wavelet Flow VAE)** (Li et al., 2024b) is a parallel work that injects the Wavelet transform into the backbone model of the VAE framework for extracting the multi-scale pyramidal features. We need to emphasize that WF-VAE introduces frequency decomposition as an inductive bias into the backbone model to simulate energy flow. In contrast, our K-Flow retains the backbone architecture and instead injects the $K$-amplitude as the realm for energy flow. In other words, K-Flow incorporates the multi-scale concept through the time domain.

**SIT (Spectral Image Tokenizer)** (Esteves et al., 2024) is a parallel work to ours that processes the spectral coefficients of input patches (image tokens) obtained through a discrete wavelet transform. Motivated by the spectral properties of natural images, SIT focuses on effectively capturing the high-frequency components of images. Furthermore, it introduces a scale-wise attention mechanism, referred to as scale-causal self-attention, which is designed to improve the model's expressiveness across multiple scales.

**DCTdiff (DCT Diffusion)** (Ning et al., 2025) explores generative modeling in the Discrete Cosine Transform (DCT) space, demonstrating that operating in the DCT domain can yield improved generation quality. While DCTdiff shares our motivation of leveraging frequency-domain transformations, it focuses specifically on the DCT representation as a fixed transform. In contrast, our $K$-amplitude framework provides a more general decomposition that encompasses DCT, wavelets, and PCA, offering greater flexibility in choosing the appropriate frequency representation for different applications.

# D  METHOD DETAILS

## D.1  FOURIER TRANSFORM AS A $K$-AMPLITUDE DECOMPOSITION

We have shown how to build the K-Amplitude through the Fourier space in Section 2.2. In the discrete setting, the Fourier transform is realized by basis functions of the form $W_N^{kn} = e^{-j\frac{2\pi}{N}kn}$, where $N$ is the length of the sequential data. An effective $K$-amplitude decomposition exploits this structure by aligning with the inherent hierarchical structure of the data manifold. For example, if most of the energy or amplitudes are concentrated in the low-scaling range, the generative capability of the flow can be enhanced by allocating more steps or resources to these low frequencies (this hypothesis is even true in the latent space, as it's demonstrated in fig. S1). Conversely, fewer steps can be allocated to high frequencies that carry minimal mass or information. For the Fourier transform, this tendency is evident in the analysis of natural images, which often exhibit the celebrated $1/f$ spectrum phenomenon (Weiss & Freeman, 2007). This phenomenon suggests that energy diminishes with increasing scaling parameter, meaning that low-scaling components hold the majority of the signal's information content.

## D.2  PCA TRANSFORM AS A $K$-AMPLITUDE DECOMPOSITION

From the $K$-amplitude perspective, PCA is an eigen-decomposition obtained by the data covariance matrix. The covariance matrix is given by:

$$\mathbf{C} = \frac{1}{n}\mathbf{X}_{\text{centered}}^\top \mathbf{X}_{\text{centered}},$$

where $\mathbf{X}_{\text{centered}} = \mathbf{X} - \overline{\mathbf{X}}$ is the centered data matrix. In this context, the principal components reveal the relative importance of each transformed direction. To translate PCA into a $K$-amplitude decomposition, we define the scaling parameter $k$ as the relative order of the principal components. For implementation, we utilize the eigenvalue decomposition of $\mathbf{C}$ for PCA, and the eigenvalues in their descending ordering define the scaling parameter $k$.

## D.3  DWT TRANSFORM AS A $K$-AMPLITUDE DECOMPOSITION

The Discrete Wavelet Transform (DWT) (Akansu & Haddad, 1992) is utilized to decompose a signal at multiple scales, capturing both time and frequency characteristics. It involves scaling and translating wavelets.

The DWT decomposes the input signal into approximation and detail coefficients:

- Given a discrete signal $x[n]$ (expressed by a finite-dimensional vector), use the scaling function $\phi(t)$ and wavelet function $\psi(t)$ to generate coefficients:

$$c_k[j] = \sum_n x[n] \cdot \phi_{k,j}[n], \qquad d_k[j] = \sum_n x[n] \cdot \psi_{k,j}[n].$$

Here, $c_k[j]$ are the approximation coefficients at scale $k$, and $d_k[j]$ are the detail coefficients at scale $k$. Comparing with our definition of K-Flow decomposition, $k$ is just a discrete scaling parameter.

The inverse transform then reconstructs the original signal from the coefficients:

$$x[n] = \sum_k c_j[k]\phi_{j,k}[n] + \sum_k d_j[k]\psi_{j,k}[n]$$

**Recursive Relationship between different Scales ($k$)** Different levels of decomposition are recursively related:

1. $k = 1$: A single level decomposition results in approximation coefficients $c_1$ and detail coefficients $d_1$;

2. $k = 2$: A two-level decomposition first produces coefficients $c_1$ and $d_1$. Then, the approximation coefficients $c_1$ are further decomposed into a second level of approximation coefficients $c_2$ and detail coefficients $d_2$.

For $k = 2$, the decomposition looks like: $x[n] \rightarrow (c_2, d_2, d_1)$, where $d_1$ represents the high-frequency components (level 1 detail coefficients) and $c_1$ is the low-frequency component (level 1 approximation coefficients). Further decomposing $c_1$ yields $c_2$ (level 2 approximation coefficients) and $d_2$ (level 2 detail coefficients). This recursive relationship illustrates why we can effectively take a finite maximum scaling parameter $k_{max}$ and still own an inverse transform.

**Practical Design choice.** In this paper's experiments, especially the Wavelet version of K-Flow flow, we take the $k_{max}$ to be one or two. One means decomposing the data into two scales, and two means decomposing the data into three scales.

**Pre-conditioning the data based on scaling** As illustrated in fig. S1, the energy distribution exhibits significant heterogeneity across different frequency bands, which consequently leads to non-uniform vector field norms in our localized K-Flow. To address this training instability, two approaches can be considered: First, following the methodology proposed in conventional diffusion models (Karras et al., 2022), we could incorporate input-output preconditioning modules into the neural architecture. However, this approach necessitates modifications to the backbone network structure, potentially affecting model compatibility and transfer learning capabilities.

In this paper, we propose a more flexible solution through component-wise normalization of the initial noise in the frequency domain. Specifically, after transforming the initial noise to the $K$-amplitude space, we normalize each discretized scaling component independently to have unit variance (var=1). This normalization is applied only to the noise, while the data remains unchanged. This approach effectively stabilizes the training dynamics by ensuring uniform noise variance across different frequency bands. More importantly, by normalizing the noise variance to unity for each scaling component, the denoising scale of the backbone model maintains consistent magnitude across different scales, which leads to more stable optimization.

### D.4 Implementation Details of K-Flow Vector Field

K-Flow is built upon a lightweight MoE (Mixture of Experts) variant of the Diffusion Transformer (DiT) (Peebles & Xie, 2023), where the experts are selected based on $k$ scales. More precisely, the integration of our method involves two key architectural modifications: (1) replacing the conventional time-embedding module with a $K$-amplitude-embedding module, where the temporal input is substituted by the scaling parameter $k$; (2) replacing the standard feedforward MLP modules with frequency-aware MoE modules that route tokens to different experts based on their $K$-amplitude components. This design enables direct incorporation of scaling information and allows the model to process different frequency bands through specialized expert networks.

For practical implementation, we provide several variants of bump functions in this subsection to facilitate exploration of the design space, with the complete training algorithm detailed in Algorithm 1. For additional insights on the $K$-amplitude localization property and its implications for model efficiency, we refer readers to Section D.5.

**Remarks.** Despite this almost model-agnostic nature, the unique $K$-amplitude localization property of Equation (11) offers an opportunity to design more efficient models. For instance, consider points that lie outside the support of function $\mathbb{I}_{\sqrt{k_x^2+k_y^2+k_z^2} \in [\lfloor k \rfloor, \lfloor k \rfloor+1)}$. In these regions, their derivative remains zero, indicating that they do not contribute to the optimization process for the corresponding scaling band. This selective activation allows us to focus computational efforts solely on the values within the support of the indicator function, $\mathbb{I}_{\sqrt{k_x^2+k_y^2+k_z^2} \in [\lfloor k \rfloor, \lfloor k \rfloor+1)}$. By doing so, the values outside this region can be treated as static conditions, providing a fixed context.

**Scaling Discretization.** In the main text, we assume, by default, that the scaling parameter $k$ takes integer values: $k \in \{0, 1, 2, \ldots, k_{max}\}$. Thus, the differentiable vector field $v_k$ for continuous $k$ is defined by interpolating between $\lfloor k \rfloor$ and $\lfloor k \rfloor + 1$.

We now extend this concept to a more general setting where $k$ may take a limited set of integer values within the range from 0 to $k_{max}$. Suppose $k_m$ and $k_n$ represent two specific integer values for $k$. We demonstrate how to extend $k$ continuously within the connected interval $[k_m, k_n]$. Let $t := k - k_m$. The differentiable version of $\phi_k$ is then expressed as:

$$\Psi_{k_m+t} = \mathcal{F}^{-1}\Bigg(\mathbb{I}_{\sqrt{k_x^2+k_y^2+k_z^2}<k_m} \cdot \mathcal{F}\{\phi\}(k_x,k_y,k_z) + \left(1 - \mathbb{I}_{\sqrt{k_x^2+k_y^2+k_z^2}\geq k_n}\right) \cdot \epsilon$$

$$+ \mathbb{I}_{\sqrt{k_x^2+k_y^2+k_z^2}\in[k_m,k_n)} \cdot (\mu(t) \cdot \mathcal{F}\{\phi\}(k_x,k_y,k_z) + (1-\mu(t)) \cdot \epsilon)\Bigg), \qquad (17)$$

where $\mu(t)$ is a bump function fulfilling $\mu(0) = \mu(k_n - k_m) = 1$ and $\mu'(0) = -\mu'(k_n - k_m)$.

Replacing the Fourier transform with the general $K$-amplitude decomposition, the K-Flow is expressed in its general form as follows:

$$\Psi_{k_m+t} = \mathcal{F}^{-1}\left(\mathbb{I}_{k<k_m} \cdot \mathcal{F}\{\phi\}(k) + \left(1 - \mathbb{I}_{\sqrt{k}\geq k_n}\right) \cdot \epsilon + \mathbb{I}_{k\in[k_m,k_n)} \cdot (\mu(t) \cdot \mathcal{F}(\{\phi\}k) + (1-\mu(t)) \cdot \epsilon)\right),$$

where $\mathcal{F}\{\phi\}(k)$ is defined in the main text.

**Experimental Implementation.** In this paper's experiments, particularly in the Fourier and PCA versions of the K-Flow flow, we restrict the discrete values of $k$ to $\{0, \frac{k_{\max}}{2}, k_{\max}\}$, with $k_{\max}$ determined by resolution. We then extend $k$ continuously using Equation 17.

**Bump Function.** We propose two types of bump functions: 1. Hard bump; 2. Soft bump. The **hard bump function** $\mu : [0,1] \to \mathbf{R}^+$ satisfies the specific endpoint properties:

$$\mu(0) = \mu(1) = 1 \quad \text{and} \quad \mu'(0) = -\mu'(1). \qquad (18)$$

Inspired by spline functions, such bump functions can be constructed using polynomials. For example, a quartic form used in our experiments is given by:

$$\mu(t) = 1 - 3t^2 + 2t^3. \qquad (19)$$

For more examples, readers can explore modifications of the connection functions used in Meyer wavelets (Meyer, 1992).

In this paper, we utilize hard bump functions for constructing K-Flow flows with scaling discretization exceeding one component.

**Soft Localization with Soft Bump Function.** Consider that the scaling parameter is discretized to take values in an increasing sequence $\{k_i\}_{i=0}^n$. Consequently, the continuous $k$ lies in the interval $k \in [k_0, k_n]$. Define

$$\psi_i := \mathbb{I}_{\sqrt{k_x^2+k_y^2+k_z^2}\in[k_i,k_{i+1})}.$$

These $\psi_i$ form a partition of unity for the K-Amplitude basis. The derivative of the soft bump function $\mu_i'$ is defined for each scaling component $\psi_i$ (a.k.a. frequency band), expressed as:

$$\mu_i'(k; a_i, b) = \begin{cases} c \cdot \left(1 - \left(\frac{k-a_i}{b}\right)^2\right)^n, & \text{if } |k - a_i| < b, \\ 0, & \text{if } |k - a_i| \geq b, \end{cases} \qquad (20)$$

where $a_i = \frac{k_i + k_{i+1}}{2}$ and $c$ is the normalization constant ensuring that the integral of the function over its compact support is 1. Note that hyper-parameter $b \leq k_n - k_0$ dictates the width or support region of the bump, while the degree $n$ measures the sharpness of the bump. We retain $b$ and $n$ as hyperparameters. The bump function $\mu_i(k)$ is then obtained by integrating $\mu_i'(k)$, which is also a polynomial function.

It is evident that $\mu_i(k)$ satisfies:

$$\mu_i(k_0) = 0 \quad \text{and} \quad \mu_i(k_n) = 1.$$

Finally, conditioned on a sampled noise $\epsilon$, the modified soft K-Flow flow at time $t \in [0, k_n - k_0]$ is expressed as:

$$\Psi_{k_0+t} = \mathcal{F}^{-1}\left(\sum_i \psi_i(k_x,k_y,k_z) \cdot \mu_i(k_0+t) \cdot \mathcal{F}\{\phi\}(k_x,k_y,k_z) + \sum_i \psi_i(k_x,k_y,k_z) \cdot (1 - \mu_i(k_0+t)) \cdot \epsilon\right).$$

$$(21)$$

Supplementary Table S2: Comparison of model parameters.

| Model | #Params (M) |
|---|---|
| VAR-B/16 | 310 |
| FlowAR-L | 590 |
| FlowAR-B (Our Impl.) | 300 |
| $K$-Flow (Fourier) (Ours) | 450 |

Through the application of this formula and a family of soft bump functions $\{\mu_i\}$, we can also implement algorithm 1. In comparison to the hard bump functions, a K-Flow constructed with soft bump functions assigns varying weights to each scale according to the scaling parameter $k$. Unlike hard bump functions which strictly set other scales to zero for each stage of $k$, soft bump functions provide a more gradual transition, allowing for multiple frequencies to flow concurrently, and the relative weights are determined by the current scaling parameter $k$.

**Comments on Haar and Meyer Wavelet $K$-amplitude.** One type of wavelet that offers both frequency and spatial localization is the Meyer wavelet. The Meyer wavelet is originally defined in the Fourier frequency domain, making it ideal for smooth frequency transitions.

The 1D Meyer wavelet $\psi(t)$ and its scaling function $\phi(t)$ are defined via their Fourier transforms, $\hat{\psi}(\omega)$ and $\hat{\phi}(\omega)$, respectively. The Meyer wavelet is constructed to ensure that the wavelet transform will partition the frequency domain into octave bands.

The Fourier transform of the scaling function $\hat{\phi}(\omega)$ is defined as:

$$\hat{\phi}(\omega) = \begin{cases} 1 & \text{if } |\omega| \le \frac{2\pi}{3}, \\ \cos\left(\frac{\pi}{2}\nu\left(\frac{3|\omega|}{2\pi}-1\right)\right) & \text{if } \frac{2\pi}{3} < |\omega| \le \frac{4\pi}{3}, \\ 0 & \text{if } |\omega| > \frac{4\pi}{3}, \end{cases} \tag{22}$$

where $\nu(t)$ is a smooth function defined as:

$$\nu(t) = \begin{cases} 0 & \text{if } t \le 0, \\ t & \text{if } 0 < t < 1, \\ 1 & \text{if } t \ge 1. \end{cases} \tag{23}$$

The Fourier transform of the Meyer wavelet $\hat{\psi}(\omega)$ is then defined as:

$$\hat{\psi}(\omega) = \begin{cases} \sin\left(\frac{\pi}{2}\nu\left(\frac{3|\omega|}{2\pi}-1\right)\right) & \text{if } \frac{2\pi}{3} < |\omega| \le \frac{4\pi}{3}, \\ 0 & \text{otherwise.} \end{cases} \tag{24}$$

In other words, Meyer transformation can be seen as the Fourier transform with a spatial cutoff window. Note that the scaling function and the wavelet function play different roles, where the low-frequency content of data is obtained by convolving the signal with the scaling function.

In the ablation section, we will employ a specific discretization of the Meyer wavelet to generate our data. Additionally, we will explore the Haar wavelet method, which is implemented solely through spatial convolution kernels and scaling operations. The Haar wavelet, being the simplest form of wavelet, is particularly interesting because it uses piecewise constant functions to capture local features at varying scales, providing a contrast to the smoother Meyer wavelet.

### D.5 IMPLEMENTATION DETAILS

**Hyper-parameters.** In our experiments, we use the pretrained VAE from Stable Diffusion (Rombach et al., 2022). The VAE encoder has a downsampling factor of 8 given an RGB pixel-based image $\mathbf{x} \in \mathbb{R}^{h \times w \times 3}$, $\mathbf{z} = \mathcal{E}(\mathbf{x})$ has shape $\frac{h}{8} \times \frac{w}{8} \times 4$. All experiments are operated in the fixed latent space.

In Table S3, we provide training hyperparameters for the image generation tasks on the two datasets. For implementing training algorithm Algorithm 1, the bump function is provided in eq. (19). For the classifier-free sampling on the conditional generation task, the cfg-scale is set to be 1.5.

Supplementary Table S3: Hyper-parameters of DiT network.

|  | **CelebA 256** | **ImageNet** |
|---|---|---|
| **Model** | DiT-L/2 (Peebles & Xie, 2023) | DiT-L/2 (Peebles & Xie, 2023) |
| **lr** | 2e-4 | 1e-4 |
| **AdamW optimizer ($\beta_1$ & $\beta_2$)** | 0.9, 0.999 | 0.9, 0.999 |
| **Batch size** | 32 | 240 |
| **# of epochs** | 500 | 900 |
| **# of GPUs** | 2 | 16 |

### D.6 Resource Requirement and Time Complexity

**Resources Requirement**. All experiments were conducted on NVIDIA H100 GPUs, with a total computational budget of approximately 3,000 GPU-hours.

**Time Complexity**. Our main focus is on comparing the computational complexity of the $K$-amplitude flow with that of ordinary latent flows, we observe that during training, the additional computational overhead introduced by the $K$-amplitude flow is minimal. From Algorithm 1, it is evident that the only additional computational step is the discrete inverse $K$-amplitude transform performed at each training iteration, while the remaining steps maintain the same complexity as the ordinary flow matching algorithm. For instance, when considering the Fourier transform, its computational complexity is $\mathcal{O}(N \log N)$, where $N$ denotes the length of the flattened image vector in the latent space.

For inference, from Algorithm 3, compared to ordinary latent flow, the only additional step to perform the $K$ amplitude flow is an inverse $K$ amplitude transform to set up the initial noise for generation, and the remaining inference remains the same complexity. Thus, we expect a similar or slightly higher complexity than the ordinary latent flow during generation.

# E  ALGORITHMS

In this section, we list three key algorithms.

---

**Algorithm 1** Training of K-Flow.

---

**Require:** Scaling parameter $k$ with maximum $k_{max}$, K-Flow transform $\mathcal{F}$, inverse transform $\mathcal{F}^{-1}$, noise distribution $p$, target distribution $q$

   Normalize $k$ to be in $[0, 1]$: $k \leftarrow k/k_{max}$
   Initialize parameters $\theta$ of $v_k$
   **while** not converged **do**
      Sample scaling parameter $k \sim \mathcal{U}(0, 1)$
      Sample training example $\phi \sim q$, sample noise $\epsilon \sim p$
      Calculate current flow position $\Psi_k$ according to K-Flow transform $\mathcal{F}$, $\mathcal{F}^{-1}$ and Equation (10)
      Calculate the conditional vector field $\dot{\Psi}_k$ according to $\mathcal{F}$, $\mathcal{F}^{-1}$ and Equation (11)
      Calculate the objective $\ell(\theta) = \|v_k(\Psi_k; \theta) - \dot{\Psi}_k\|_g^2$, following Equation (12)
      $\theta = \text{optimizer\_step}(\ell(\theta))$
   **end while**

---

**Algorithm 2** Scaling-controllable Generation of K-Flow.

---

**Require:** Scaling parameter $k$, $K$-amplitude transform $\mathcal{F}$, inverse transform $\mathcal{F}^{-1}$, noise distribution $p$ in the $K$-amplitude space, target distribution $q$

   Initialize pre-trained $v_k(\theta)$
   Sample one high-scaling noise $\epsilon_{\text{high}} \sim p$, sample two independent low-scaling noise $\epsilon_{\text{low}} \sim p$, $\tilde{\epsilon}_{\text{low}} \sim p$
   $\Psi_0 = \mathcal{F}^{-1}\{\epsilon_{\text{low}}, \epsilon_{\text{high}}\}$
   $\tilde{\Psi}_0 = \mathcal{F}^{-1}\{\tilde{\epsilon}_{\text{low}}, \epsilon_{\text{high}}\}$
   **for** $k \in [0, 1]$ **do**
      $\Psi_k \leftarrow \text{ODEstep}(v_k(\cdot, \theta), \Psi_0)$
      $\tilde{\Psi}_k \leftarrow \text{ODEstep}(v_k(\cdot, \theta), \tilde{\Psi}_0)$
   **end for**

   **return** $\Psi_1, \tilde{\Psi}_1$

---

**Algorithm 3** Class-conditional Generation of K-Flow with dropping.

---

**Require:** Pre-trained $v_k(\theta)$, conditioning class $c$, dropping time $\tau$, noise distribution $p$, guidance parameter $\omega$

1: $\Psi_0 \sim p$
2: **for** $k \in [0, \tau]$ **do**
3:    $\tilde{v}_k(\cdot) \leftarrow (1 - \omega)v_k^\theta(\cdot, \theta) + \omega u_k(\cdot, c, \theta)$   {guided velocity}
4:    $\Psi_\tau \leftarrow \text{ODEstep}(\tilde{v}_k(\cdot), \Psi_0)$
5: **end for**
6: **for** $k \in [\tau, 1]$ **do**
7:    $\Psi_1 \leftarrow \text{ODEstep}(v_k(\cdot, \theta), \Psi_\tau)$
8: **end for**
9:
10: **return** $\Psi_1$

---

## F    MORE RESULTS

**Discussion on the quantitative generation results**    We posit that standard generative modeling metrics, such as FID and Recall, may not fully capture the specific advantages conferred by K-Flow, particularly concerning improvements in structural fidelity across the full spectrum of K-amplitudes. The benefits of our method become more apparent in downstream applications where high-fidelity detail is critical. For instance, as demonstrated in Appendix F.9, K-Flow exhibits stronger performance on tasks like super-resolution and molecular assembly. This superior performance on detail-dependent tasks supports the conclusion that K-Flow offers tangible advantages in generating high-fidelity structures, benefits that may not be fully reflected by conventional, perception-oriented metrics alone.

**Comments on performance.**    Firstly, one of our primary motivations is indeed to build upon the observation that some generative models, like DDPMs, appear to learn a process that progresses from low to high frequencies. However, we note that this property is often an empirical post-hoc observation, tested with specific metrics, and its existence and characteristics within the flow matching paradigm are not well understood. From a PCA perspective, a standard flow model fails to quantitatively exhibit the low-to-high frequency hierarchy. Thus, the contribution of our work is to transition this from an implicit, emergent property to an explicit and structured design principle. K-Flow provides a generalized framework that formalizes this frequency-aware path, offering fine-grained control that is absent in standard models. Secondly, standard metrics like FID and Recall may not fully capture the benefits of improved structural fidelity across all K-amplitudes.

### F.1    MORE RESULTS ON UNCONDITIONAL GENERATION

We provide more results on the class-conditional generation using K-Flow in Figure S4.

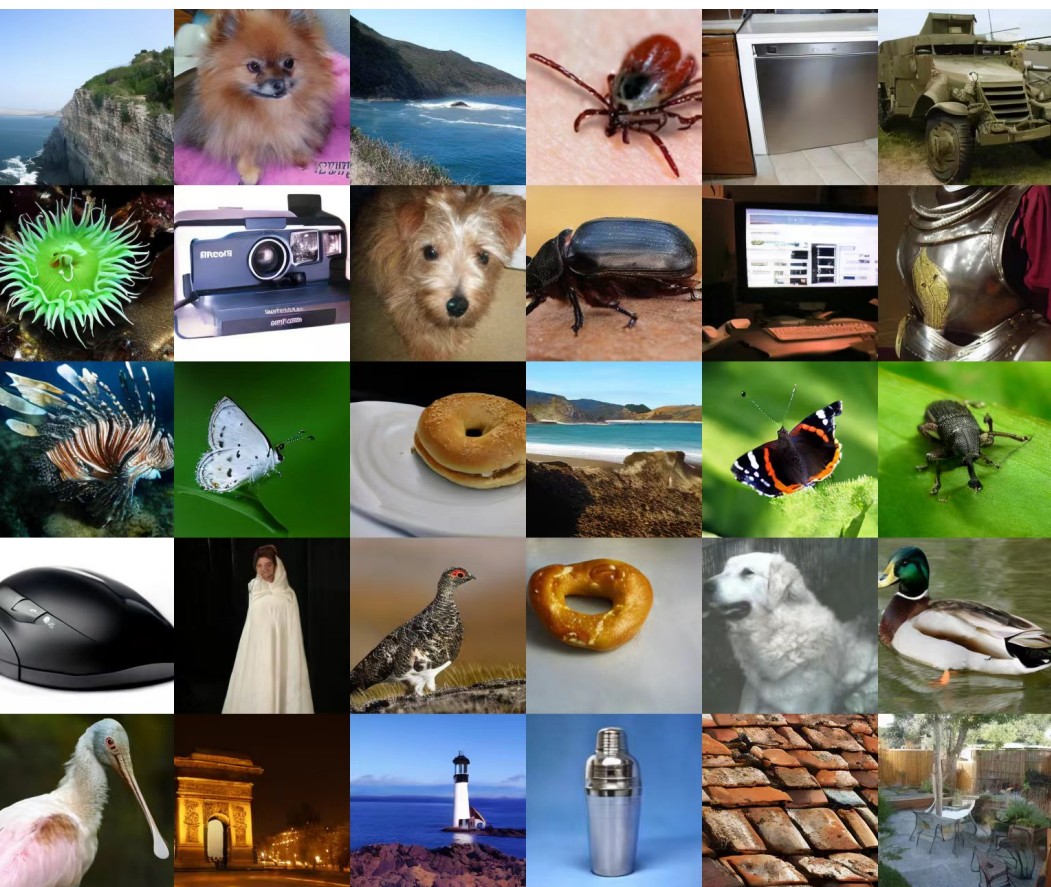

Supplementary Figure S4: Non-curated samples of our reversing scaling variant on ImageNet (cfg = 1.5).

## F.2 CLASS-AWARE FID METRIC

We propose using the class-aware FID metric, defined as follows:

$$\text{FID}_{\text{class-conditional}} = \mathbb{E}_{c \sim p(c)} \left[ \text{FID}(c) \right] \tag{25}$$

where for each class $c$, the FID is calculated by:

$$\text{FID}(c) := \text{FID}(X_r^c, X_g^c) = \|\mu_r^c - \mu_g^c\|^2 + \text{Tr}(\Sigma_r^c + \Sigma_g^c - 2(\Sigma_r^c \Sigma_g^c)^{1/2}). \tag{26}$$

Here, $X_r^c$ and $X_g^c$ denote the real and generated data subsets for class $c$, respectively. Based on FID$(c)$, the Class-Dropping-Ratio (CDR) is defined by

$$\text{CDR} := \mathbb{E}_{c \sim p(c)} \left[ \frac{\text{FID}_{\text{aft}}(c)}{\text{FID}_{\text{bef}}(c)} \right]$$

where $\text{FID}_{\text{bef}}$ denotes the FID calculated for the flow model carried with the class condition for the whole process, and $\text{FID}_{\text{aft}}$ denotes the FID calculated for the flow model carried with the class condition for only a subprocess (we keep the initial 30% of the inference time for the experiment). In practice, instead of computing the expectation over the entire class distribution $p(c)$, we randomly select 5 classes out of the total 1000 classes for evaluation.

## F.3 ABLATION STUDIES ON CONTROLLABLE CLASS-CONDITIONAL GENERATION

In Section 4, we provide brief results on the controllable class-conditional generation over ImageNet. Here, we would like to give a more qualitative comparison between our model K-Flow and LFM.

**K-Flow, Undrop**          **K-Flow, Drop**

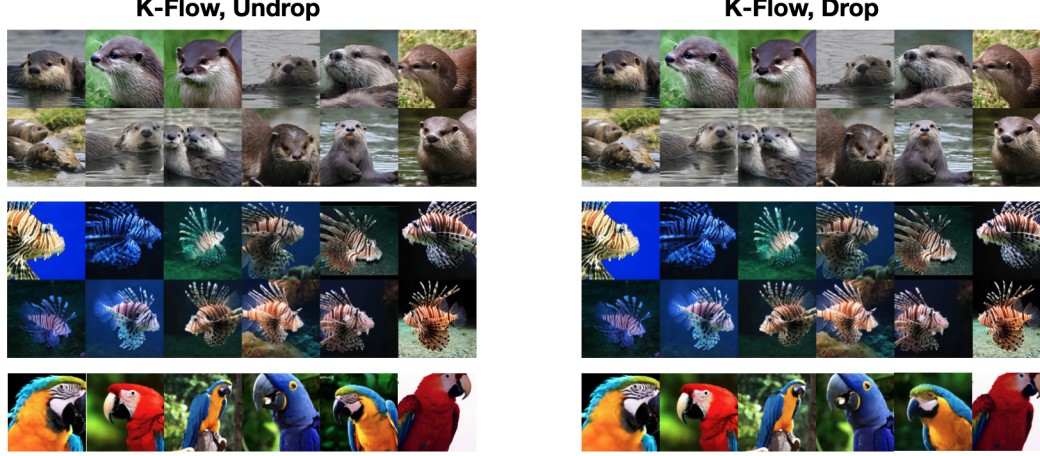

Supplementary Figure S5: Classifier-free guidance sampling of our Fourier-based K-Flow with a hyperparameter setting of cfg = 3. In the right columns, the class condition is omitted for the last 50% of the scaling steps during inference, using the same initial noise. It can be observed that as the cfg value increases and the duration of omitting the class condition decreases, the generated results appear nearly identical.

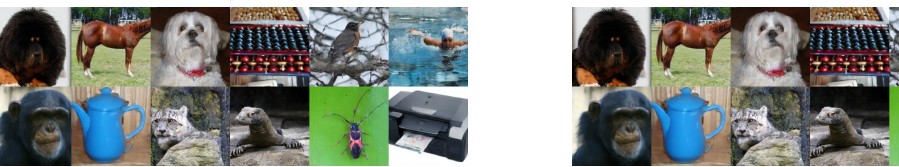

Supplementary Figure S6: Classifier-free guidance sampling of our wavelet-based K-Flow with a hyperparameter setting of cfg = 2. In the right columns, the class condition is drooped for the last 70% of the scaling steps during inference, using the same initial noise. It can be observed that after dropping, K-Flow still preserves the high-scaling contents.

## F.4 ABLATION STUDY ON WAVELET BASE

From Table S4, we tested two additional wavelet base, the discrete Haar basis (Haar, 1911) and the discrete Myer basis (Meyer, 1990) as a supplement of the Daubechies wavelet (db6, Karam (2012)) used in the main text. All three wavelets demonstrated comparable performance in terms of both the FID and Recall metrics.

Supplementary Table S4: CelebA-HQ 256.

| Model | FID↓ | Recall↑ |
|---|---|---|
| **CelebA-HQ 256** | | |
| K-Flow, Meyer-DiT L/2 | **5.01** | **0.47** |
| K-Flow, Haar-DiT L/2 | 5.01 | 0.46 |
| K-Flow, Db-DiT L/2 (three scales) | 5.77 | 0.42 |

Supplementary Table S5: Conditional ImageNet 256.

| Model | FID↓ | Recall↑ |
|---|---|---|
| K-Flow, Wave (reverse) | 23.06 | 0.58 |
| + cfg=1.5 | 5.10 | 0.46 |
| K-Flow, Wave-DiT L/2 (Ours) | 17.8 | 0.56 |
| + cfg=1.5 | 4.49 | 0.44 |
| K-Flow, Wave-DiT L/2 (three bands) | 16.1 | - |
| LFM, DiT L/2 (Dao et al., 2023) | 14.0 | 0.56 |
| + cfg=1.5 | 2.85 | 0.42 |

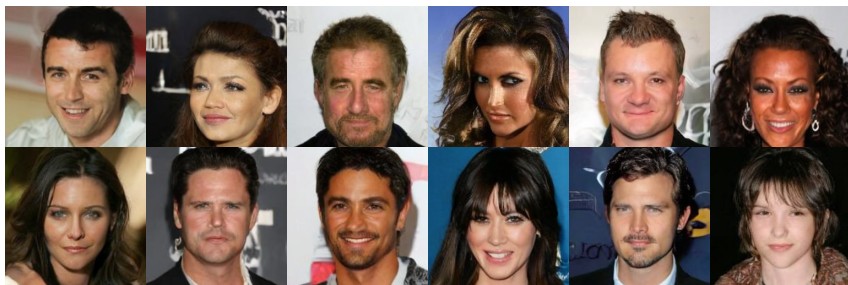

Supplementary Figure S7: Daubechies wavelet $K$-amplitude with more components trained on CelebA-256.

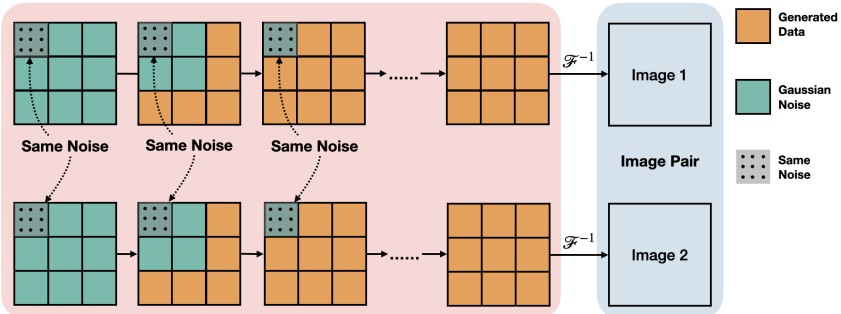

Supplementary Figure S8: Pipeline of scaling-controllable generation (high scaling).

### F.5 ABLATION STUDY ON SCALING PARTITIONS

Although the quality of face generation appears similar to the naked eye, the model with three $K$-amplitude bands (the last row of Table S4) performed worse in terms of FID and Recall metrics. We provide the generated samples for qualitative evaluation in Figure S7.

**Reversing the $K$-amplitude Scaling.** In Table S5, we also tested a counterintuitive scaling order: from high to low. This means generating high-frequency details first and then filling in the low-frequency components during the generation process. We find that the model can still produce images normally (Figure S4), with a better diversity (Recall) but lower quality (FID) compared to the low-to-high scaling approach.

### F.6 IMAGE SCALING-CONTROLLABLE GENERATION

**Preserving Low Scaling, Modifying High Scaling.** We need to highlight that in K-Flow, when modeling the flow from lower to higher scales, the noise at higher scales is used to predict the velocity at the lower scale. This is determined by the nature of ODE flow. To this end, we conduct a study by reversing the scaling direction in the Daubechies wavelet K-Flow, and the pipeline is illustrated in Figure S8. In such a reversed setup, we keep the low-scaling part the same noise while gradually denoising the high-scaling part.

The results are listed in Figure S9. According to the six pairs of results, we can observe that the low-scaling part stays the same, like the background of the image and the gender and color of the people, while the high-resolution details of facial expressions and outlook vary within each pair.

**Remarks.** Although the overall results are generally optimistic, some unexpected changes have been observed in the high-scaling parts. This may be attributed to two factors:

1. The compressed latent space may mix high and low content present in the original pixel space.
2. The loss Equation (12) may not be perfectly optimized, meaning that K-Flow localized vector field might not be perfectly confined to the low-scaling part. The second factor might be mitigated by training on larger datasets. Furthermore, by training a reversed K-Flow flow (from high to low), we observe that fixing the low-scaling noise enables unsupervised editing of detailed high-scaling content.

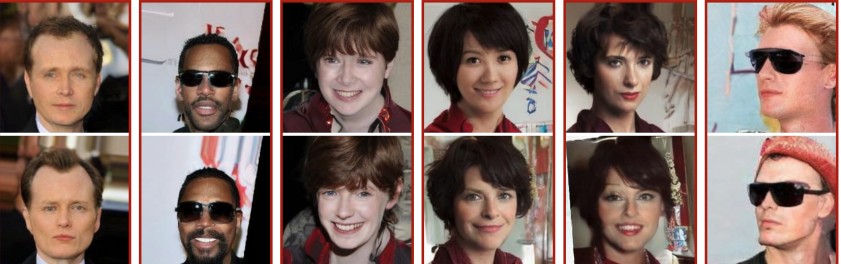

Supplementary Figure S9: Results of scaling-controllable generation. We display six pairs of images, where each pair of images preserves the low scaling and differs in the high scaling.

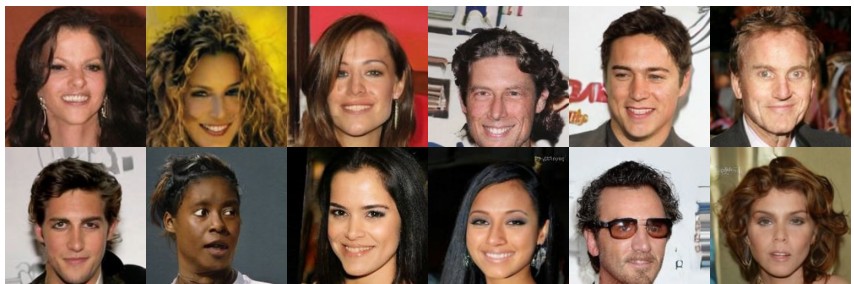

Supplementary Figure S10: LFM editing by Algorithm 2.

In Figure S9, we've tested the wavelet-based K-Flow and observed similar results with the Fourier-based K-Flow.

This insight further supports our model's capacity to decompose the generative process into distinct frequency bands, where specific frequency bands can be independently controlled. This separation aids in achieving more detailed and deliberate modifications to generated data, adding a layer of precision and flexibility to the generative framework.

## F.7   IMAGE RESTORATION

In this section, we evaluate the performance of the $K$-amplitude flow on image restoration tasks, including super-resolution and deblurring. These tasks typically involve reconstructing the high-frequency components of an image conditioned on the known low-frequency components. Unlike unsupervised editing based on different scales, the performance of this experiment can be quantitatively measured using reconstruction metrics such as Peak Signal-to-Noise Ratio (PSNR) and Structural Similarity (SSIM).

**Datasets and Baselines**   Our method is benchmarked against standard diffusion and flow matching based restoration methods (see (Martin et al., 2024) for a detailed introduction). We evaluate all methods on CelebA dataset, with images resized to $128 \times 128$.

**Algorithm.** Our training-free restoration method basically follows the efficient algorithm proposed in Martin et al. (2024) with two key changes adapted to $K$-amplitude:

1. The naive linear interpolation step is replaced by our scaling interpolation formula Equation (10).
2. Instead of starting restoration from pure noise, we start at $t = 0.5$, since our flow primarily denoises high-frequency components during the later period of time. From the $K$-amplitude perspective, this strategic initialization point provides a more informed starting state compared to conventional flow approaches while maintaining restoration quality.

We provide the algorithm details in Algorithm 4.

Supplementary Table S6: Performance comparison on image restoration. In this experiment, we pre-trained $K$ flow with the same U-net architecture implemented (Martin et al., 2024).

| | Super-res. | | Deblurring. | |
|---|---|---|---|---|
| | PSNR | SSIM | PSNR | SSIM |
| Degraded | 10.17 | 0.182 | 27.67 | 0.740 |
| PnP-Diff (Zhu et al., 2023) | 31.20 | 0.893 | 32.49 | 0.911 |
| PnP-GS (Hurault et al., 2021) | 30.69 | 0.889 | 33.65 | 0.924 |
| OT-ODE Pokle et al. (2023) | 31.05 | 0.902 | 32.63 | 0.915 |
| D-Flow (Ben-Hamu et al., 2024) | 29.17 | 0.833 | 31.07 | 0.877 |
| Flow-Priors Zhang et al. (2024) | 28.35 | 0.717 | 31.40 | 0.856 |
| PnP-Flow (Martin et al., 2024) | 31.49 | 0.907 | 34.51 | 0.940 |
| K-Flow (ours) | **32.35** | **0.928** | **35.72** | **0.942** |

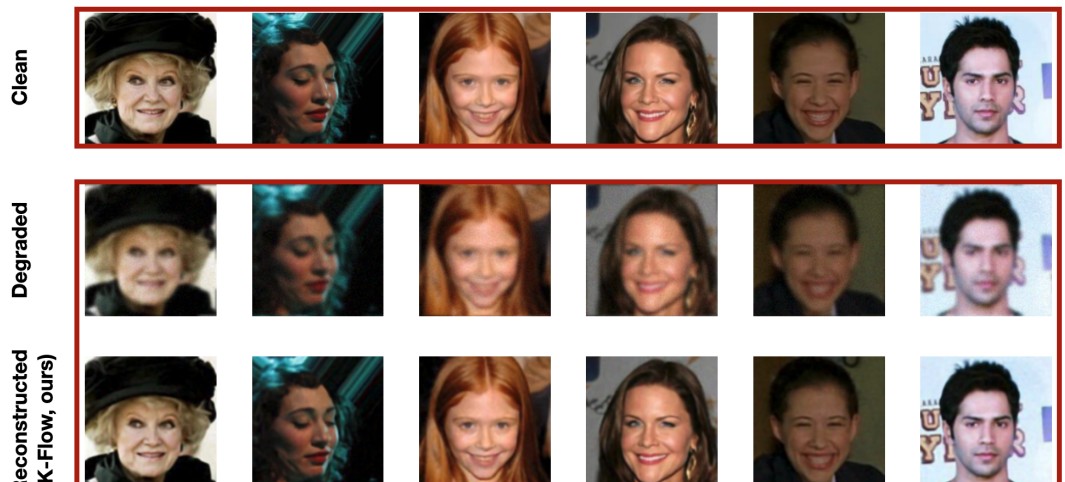

Supplementary Figure S11: Visualization for image restoration using K-Flow.

---

**Algorithm 4** PnP K-Flow.

---

**Input:** Pre-trained network $v^\vartheta$ by K-Flow, time sequence $(t_n)_n$ either finite with
$\quad t_n = n/N$, $N \in \mathbb{N}$ or infinite with $\lim_{n \to +\infty} t_n = 1$ and $t_0 = 0.3$, adaptive stepsizes $(\gamma_n)_n$.
**Initialize:** $x_0 \in \mathbb{R}^d$.
**for** $n = 0, 1, \ldots,$ **do**
$\quad z_n = x_n - \gamma_n \nabla F(x_n)$. $\qquad\qquad\qquad$ ▷ Gradient step on the data-fidelity term
$\quad \tilde{z}_n$ from $z_n$ and noise $\epsilon$ through $K$-amplitude interpolation 10.
$\quad x_{n+1} = D_{t_n}(\tilde{z}_n)$ $\qquad\qquad\qquad$ ▷ PnP step with restoration denoiser in (Martin et al., 2024)
**return** $x_{n+1}$

---

**Results.** We report benchmark results (following Martin et al. (2024)) for all methods on super-resolution and deblurring tasks, measuring average PSNR and SSIM on 100 test images, including super-resolution (with down sample rate ×2). Results are averaged across 100 test images. From Table S6, we see that the $K$-amplitude flow achieves state-of-the-art (SOTA) quantitative results in both the super-resolution and deblurring tasks. This superior performance without task-specific hyperparameter tuning can be attributed to our model's inherent frequency-aware design: both super-resolution and deblurring tasks primarily involve recovering high-frequency information (higher values of the scaling parameter $k$), which naturally aligns with the later stages of K-Flow's scaling-progressive generation process. From Figure S11, we can clearly see how K-Flow restores the high scaling components.

Supplementary Table S7: K-Flow against seven generative models on COD-Cluster17 with 5K, 10K, and all samples. The best results are marked in **bold**.

| | COD-Cluster17-5K | | COD-Cluster17-10K | | COD-Cluster17-All | |
|---|---|---|---|---|---|---|
| | PM (atom) ↓ | PM (center) ↓ | PM (atom) ↓ | PM (center) ↓ | PM (atom) ↓ | PM (center) ↓ |
| GNN-MD | $13.67 \pm 0.06$ | $13.80 \pm 0.07$ | $13.83 \pm 0.06$ | $13.90 \pm 0.05$ | $22.30 \pm 12.04$ | $14.51 \pm 0.82$ |
| CrystalSDE-VE | $15.52 \pm 1.48$ | $16.46 \pm 0.99$ | $17.25 \pm 2.46$ | $17.86 \pm 1.11$ | $17.28 \pm 0.73$ | $18.92 \pm 0.03$ |
| CrystalSDE-VP | $18.15 \pm 3.02$ | $19.15 \pm 4.46$ | $22.20 \pm 3.29$ | $21.39 \pm 1.50$ | $18.03 \pm 4.56$ | $20.02 \pm 3.70$ |
| CrystalFlow-VE | $14.87 \pm 7.07$ | $13.08 \pm 4.51$ | $16.41 \pm 2.64$ | $16.71 \pm 2.35$ | $12.80 \pm 1.20$ | $15.09 \pm 0.34$ |
| CrystalFlow-VP | $15.71 \pm 2.69$ | $17.10 \pm 1.89$ | $19.39 \pm 4.37$ | $16.01 \pm 3.13$ | $13.50 \pm 0.44$ | $13.28 \pm 0.48$ |
| CrystalFlow-LERP | $13.59 \pm 0.09$ | $13.26 \pm 0.09$ | $13.54 \pm 0.03$ | $13.20 \pm 0.03$ | $13.61 \pm 0.00$ | $13.28 \pm 0.01$ |
| AssembleFlow | $7.27 \pm 0.04$ | $6.13 \pm 0.10$ | $7.38 \pm 0.03$ | $6.21 \pm 0.05$ | $7.37 \pm 0.01$ | $6.21 \pm 0.01$ |
| K-Flow (ours) | $\mathbf{7.21 \pm 0.12}$ | $\mathbf{6.11 \pm 0.11}$ | $\mathbf{7.26 \pm 0.06}$ | $\mathbf{6.12 \pm 0.07}$ | $\mathbf{7.23 \pm 0.01}$ | $\mathbf{6.07 \pm 0.01}$ |

## F.8 MOLECULAR ASSEMBLY

We consider another scientific task: molecular assembly. The goal is to learn the trajectory on moving clusters of weakly-correlated molecular structures to the strongly-correlated structures.

**Dataset and evaluation metrics.** We evaluate our method using the crystallization dataset COD-Cluster17 (Liu et al., 2024b), a curated subset of the Crystallography Open Database (COD)(Grazulis et al., 2009) containing 133K crystals. We consider three versions of COD-Cluster17 with 5K, 10K, and the full dataset. To assess the quality of the generated molecular assemblies, we employ *Packing Matching (PM)*(Chisholm & Motherwell, 2005), which quantifies how well the generated structures align with reference crystals in terms of spatial arrangement and packing density. Following (Liu et al., 2024b), we compute PM at both the atomic level (PM-atom) and the mass-center level (PM-center) (Chisholm & Motherwell, 2005).

**Baselines.** We evaluate our approach against GNN-MD (Liu et al., 2024b), variations of CrystalSDE and CrystalFlow (Liu et al., 2024b), and the state-of-the-art AssembleFlow (Guo et al., 2025). CrystalSDE-VE/VP model diffusion via stochastic differential equations, while CrystalFlow-VE/VP use flow matching, with VP focusing on variance preservation. CrystalFlow-LERP employs linear interpolation for efficiency. AssembleFlow (Guo et al., 2025) enhances rigidity modeling using an inertial frame.

**Main results.** The main results in Table S7 show that K-Flow outperforms all baselines across three datasets. Building on AssembleFlow's rigidity modeling, K-Flow decomposes molecular pairwise distances via spectral methods and projects geometric information from $\mathbb{R}^3$ and $\mathrm{SO}^3$ accordingly. This approach achieves consistently superior packing matching performance.

