**Image Restoration.** While traditional semantic metrics cannot directly assess unsupervised frequency-based editing, we quantitatively validate our scaling-aware generation path via image restoration tasks, where frequency-specific changes are objectively measurable. This includes super-resolution, inpainting, and deblurring. From Table S8 in Appendix, K-Flow achieves state-of-the-art performance in terms of PSNR and SSIM metrics on the CelebA dataset, while requiring only half the iterations compared to the baseline PnP-flow (Martin et al., 2024). This demonstrates that our frequency-domain formulation enables more efficient and accurate high-frequency reconstruction. Detailed experimental settings and algorithms are provided in Appendix F.

## 5 CONCLUSION

In this paper, we introduce K-Flow Matching (K-Flow), an efficient flow-matching model that flows along the $K$-amplitude for generative modeling. K-Flow naturally generalizes the multi-scales of data (*e.g.*, multi-resolution or frequencies in images) to multi-scales in the $K$-amplitude space.

**Future Directions.** As we have verified the effectiveness of K-Flow exclusively on image generation tasks, moving forward, two promising directions are worth exploring. (1) Multimodal generation: This includes tasks such as large-scale data generation guided by dense captions, which could better showcase the steerability of K-Flow by aligning images with natural language inputs. (2) We outline six properties of K-Flow in Section 1, *e.g.*, the amplitude naturally corresponds to energy. While Section 3 briefly discusses how energy is represented in K-Flow, this aspect has not been explored in depth. We believe that such energy term holds potential for integration with the utility of energy-based models in future work.

## ETHICS STATEMENT

This research adheres fully to the ICLR Code of Ethics. It does not involve human subjects, personal information, or sensitive data. All datasets and code used or released comply with their respective licenses and terms of use. The contributions of this work are methodological and foundational, raising no concerns related to fairness, privacy, security, or potential misuse.

## REPRODUCIBILITY STATEMENT

We are committed to ensuring reproducibility of our results. Comprehensive details, including dataset access, experimental setup, model configurations, evaluation metrics, and checkpoints, will be made publicly available on GitHub upon acceptance. Clear documentation and scripts will be provided to enable accurate replication of all main results, supporting transparency and scientific rigor.

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

multi-scale decomposed data. Specifically, after performing $K$-amplitude decomposition, we compute the statistical moments (mean and standard deviation) for each discretized scaling component independently. This normalized representation is then processed through the flow, followed by an inverse normalization step to restore the original scale. This pre-processing approach effectively stabilizes the training dynamics while preserving the architectural integrity of the backbone model.

### D.4 IMPLEMENTATION DETAILS OF K-FLOW VECTOR FIELD

K-Flow is architecture-agnostic in terms of its vector field implementation, making it compatible with classical architectures such as U-Net (Song et al., 2020) and Vision Transformers (Peebles & Xie, 2023) that are widely adopted in continuous normalizing flows and diffusion models. The integration of our method requires only one targeted modification: replacing the conventional time-embedding module with a $K$-amplitude-embedding module, where the temporal input is substituted by the scaling parameter $k$. This modification enables direct incorporation of scaling information while preserving the original architectural benefits, though we leave the exploration of specialized architectures for $K$-amplitude flow as future work.

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

Supplementary Table S3: CelebA-HQ 256.

| Model | NFE ↓ |
|---|---|
| LFM, ADM | 85 |
| LFM, DiT L/2 | 89 |
| FM | 128 |
| K-Flow, DiT L/2 (Ours) | 78 |

It is worth mentioning that when testing the FID, we apply the fixed-step ODE solver ("Euler") with 50 steps. Thus, we also provide the average inference time of generating one CelebA sample on one H20 GPU:

Supplementary Table S4: CelebA-HQ 256.

| Model | Time (s) |
|---|---|
| LFM, DiT L/2 | 0.583 |
| K-Flow, DiT L/2 (Ours) | 0.589 |

## E  ALGORITHMS

In this section, we list three key algorithms.

---

**Algorithm 1** Training of K-Flow.

---

**Require:** Scaling parameter $k$ with maximum $k_{max}$, K-Flow transform $\mathcal{F}$, inverse transform $\mathcal{F}^{-1}$, noise distribution $p$, target distribution $q$
    Normalize $k$ to be in $[0, 1]$: $k \leftarrow k/k_{max}$
    Initialize parameters $\theta$ of $v_k$
    **while** not converged **do**
        Sample scaling parameter $k \sim \mathcal{U}(0, 1)$
        Sample training example $\phi \sim q$, sample noise $\epsilon \sim p$
        Calculate current flow position $\Psi_k$ according to K-Flow transform $\mathcal{F}$, $\mathcal{F}^{-1}$ and Equation (10)
        Calculate the conditional vector field $\dot{\Psi}_k$ according to $\mathcal{F}$, $\mathcal{F}^{-1}$ and Equation (11)
        Calculate the objective $\ell(\theta) = \|v_k(\Psi_k; \theta) - \dot{\Psi}_k\|_g^2$, following Equation (12)
        $\theta = \text{optimizer\_step}(\ell(\theta))$
    **end while**

---

**Algorithm 2** Scaling-controllable Generation of K-Flow.

---

**Require:** Scaling parameter $k$, $K$-amplitude transform $\mathcal{F}$, inverse transform $\mathcal{F}^{-1}$, noise distribution $p$ in the $K$-amplitude space, target distribution $q$
    Initialize pre-trained $v_k(\theta)$
    Sample one high-scaling noise $\epsilon_{\text{high}} \sim p$, sample two independent low-scaling noise $\epsilon_{\text{low}} \sim p$, $\tilde{\epsilon}_{\text{low}} \sim p$
    $\Psi_0 = \mathcal{F}^{-1}\{\epsilon_{\text{low}}, \epsilon_{\text{high}}\}$
    $\tilde{\Psi}_0 = \mathcal{F}^{-1}\{\tilde{\epsilon}_{\text{low}}, \epsilon_{\text{high}}\}$
    **for** $k \in [0, 1]$ **do**
        $\Psi_k \leftarrow \text{ODEstep}(v_k(\cdot, \theta), \Psi_0)$
        $\tilde{\Psi}_k \leftarrow \text{ODEstep}(v_k(\cdot, \theta), \tilde{\Psi}_0)$
    **end for**

    **return**  $\Psi_1, \tilde{\Psi}_1$

---

**Algorithm 3** Class-conditional Generation of K-Flow with dropping.

---

**Require:** Pre-trained $v_k(\theta)$, conditioning class $c$, dropping time $\tau$, noise distribution $p$, guidance parameter $\omega$
  1: $\Psi_0 \sim p$
  2: **for** $k \in [0, \tau]$ **do**
  3:    $\tilde{v}_k(\cdot) \leftarrow (1 - \omega)v_k^\theta(\cdot, \theta) + \omega u_k(\cdot, c, \theta)$  {guided velocity}
  4:    $\Psi_\tau \leftarrow \text{ODEstep}(\tilde{v}_k(\cdot), \Psi_0)$
  5: **end for**
  6: **for** $k \in [\tau, 1]$ **do**
  7:    $\Psi_1 \leftarrow \text{ODEstep}(v_k(\cdot, \theta), \Psi_\tau)$
  8: **end for**
  9:
10: **return**  $\Psi_1$

---

# F MORE RESULTS

## F.1 MORE RESULTS ON UNCONDITIONAL GENERATION

We provide more results on the class-conditional generation using K-Flow in Figure S4.

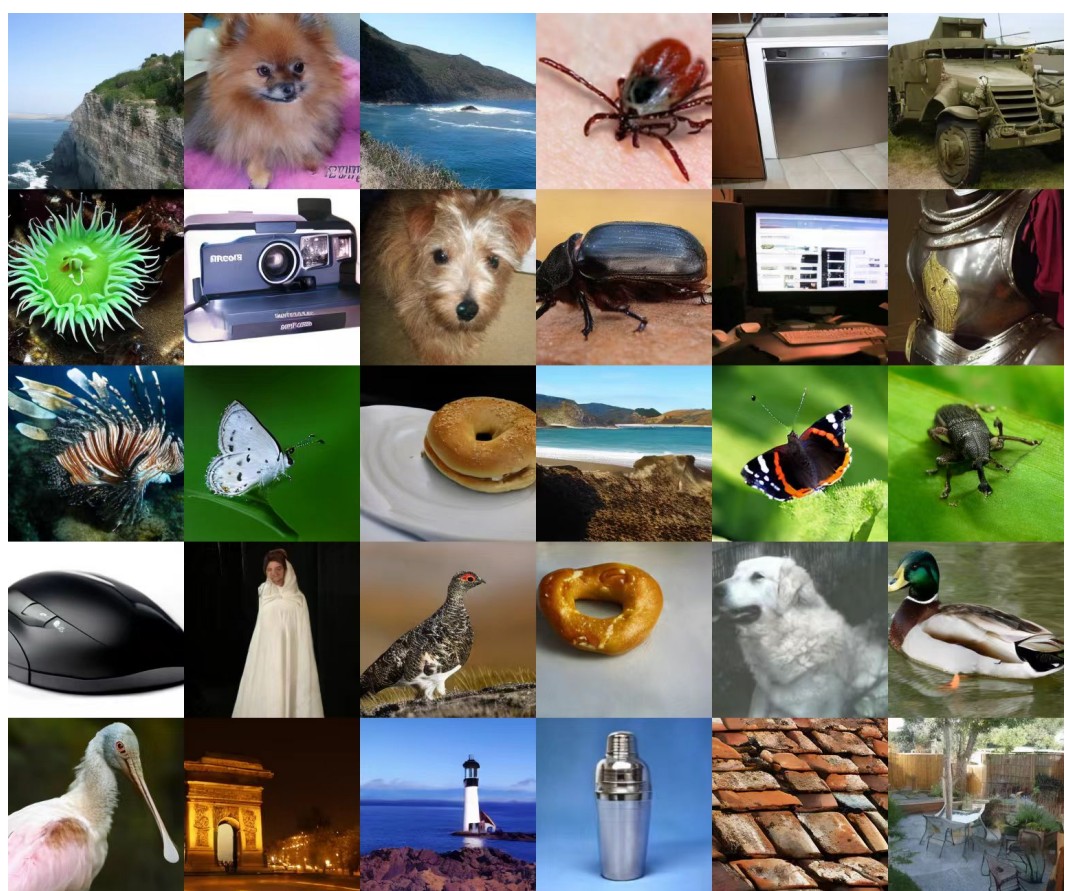

Supplementary Figure S4: Non-curated samples of our reversing scaling variant on ImageNet (cfg = 1.5).

## F.2 Unconditional Generation on LSUN Church

We conducted unconditional generation experiments on LSUN Church Yu et al. (2015), with the resolution of $256 \times 256$. The results are presented in Table S5. We test our $K$-amplitude flow with two and three scaling components using the db6 wavelet Karam (2012) $K$-amplitude transform, and we find that the three scaling components version achieves the best quantitative results in terms of FID and Recall.

Supplementary Table S5: LSUN Church $256 \times 256$.

| Model | FID ↓ | Recall ↑ |
|---|---|---|
| LFM (ADM) | 7.7 | 0.39 |
| LFM (DiT L/2) | 5.54 | 0.48 |
| FM | 10.54 | - |
| LDM | 4.02 | 0.52 |
| WaveDiff | 5.06 | 0.40 |
| DDPM | 7.89 | - |
| ImageBART | 7.32 | - |
| K-Flow, two scales (Ours) | 5.37 | 0.47 |
| K-Flow, three scales (Ours) | **5.19** | **0.49** |

**Results**  Table S5 summarizes the results on LSUN Church. We test our K-Flow with two and three scaling components using the db6 wavelet $K$-amplitude transform, and we find that the three scaling components version achieves the best quantitative results in terms of FID and Recall.

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

We provide the algorithm details in Algorithm 4.

---

**Algorithm 4** PnP K-Flow.

---

**Input:** Pre-trained network $v^\vartheta$ by K-Flow, time sequence $(t_n)_n$ either finite with
  $t_n = n/N$, $N \in \mathbb{N}$ or infinite with $\lim_{n \to +\infty} t_n = 1$ and $t_0 = 0.3$, adaptive stepsizes $(\gamma_n)_n$.
**Initialize:** $x_0 \in \mathbb{R}^d$.
**for** $n = 0, 1, \ldots,$ **do**
  $z_n = x_n - \gamma_n \nabla F(x_n)$.                    ▷ Gradient step on the data-fidelity term
  $\tilde{z}_n$ from $z_n$ and noise $\epsilon$ through $K$-amplitude interpolation 10.
  $x_{n+1} = D_{t_n}(\tilde{z}_n)$                    ▷ PnP step with restoration denoiser in (Martin
et al., 2024)
**return** $x_{n+1}$

---

**Results.** We report benchmark results (following Martin et al. (2024)) for all methods across three restoration tasks, measuring average PSNR and SSIM on 100 test images, including super-resolution (with down sample rate $\times 2$), deblurring and Box inpainting problems. Results are averaged across 100 test images. From Table S8, we see that the $K$-amplitude flow achieves state-of-the-art (SOTA) quantitative results in the super-resolution task, deblurring task, and comparable results in inpainting tasks. In terms of time complexity, we only use 75 iterations in the super-resolution task, while PnP-flow's iteration number is set to 150. This superior performance without task-specific hyperparameter tuning can be attributed to our model's inherent frequency-aware design: both deblurring and super-resolution tasks primarily involve recovering high-frequency information (higher values of the scaling parameter $k$ ), which naturally aligns with the later stages of K-Flow's scaling-progressive generation process. From Figure S11, we can clearly see how K-Flow restores the high scaling components of a blurred picture.

### F.9 MOLECULAR ASSEMBLY

We consider another scientific task: molecular assembly. The goal is to learn the trajectory on moving clusters of weakly-correlated molecular structures to the strongly-correlated structures.

Supplementary Table S9: K-Flow against seven generative models on COD-Cluster17 with 5K, 10K, and all samples. The best results are marked in **bold**.

| | COD-Cluster17-5K | | COD-Cluster17-10K | | COD-Cluster17-All | |
|---|---|---|---|---|---|---|
| | PM (atom) $\downarrow$ | PM (center) $\downarrow$ | PM (atom) $\downarrow$ | PM (center) $\downarrow$ | PM (atom) $\downarrow$ | PM (center) $\downarrow$ |
| GNN-MD | $13.67 \pm 0.06$ | $13.80 \pm 0.07$ | $13.83 \pm 0.06$ | $13.90 \pm 0.05$ | $22.30 \pm 12.04$ | $14.51 \pm 0.82$ |
| CrystalSDE-VE | $15.52 \pm 1.48$ | $16.46 \pm 0.99$ | $17.25 \pm 2.46$ | $17.86 \pm 1.11$ | $17.28 \pm 0.73$ | $18.92 \pm 0.03$ |
| CrystalSDE-VP | $18.15 \pm 3.02$ | $19.15 \pm 4.46$ | $22.20 \pm 3.29$ | $21.39 \pm 1.50$ | $18.03 \pm 4.56$ | $20.02 \pm 3.70$ |
| CrystalFlow-VE | $14.87 \pm 7.07$ | $13.08 \pm 4.51$ | $16.41 \pm 2.64$ | $16.71 \pm 2.35$ | $12.80 \pm 1.20$ | $15.09 \pm 0.34$ |
| CrystalFlow-VP | $15.71 \pm 2.69$ | $17.10 \pm 1.89$ | $19.39 \pm 4.37$ | $16.01 \pm 3.13$ | $13.50 \pm 0.44$ | $13.28 \pm 0.48$ |
| CrystalFlow-LERP | $13.59 \pm 0.09$ | $13.26 \pm 0.09$ | $13.54 \pm 0.03$ | $13.20 \pm 0.03$ | $13.61 \pm 0.00$ | $13.28 \pm 0.01$ |
| AssembleFlow | $7.27 \pm 0.04$ | $6.13 \pm 0.10$ | $7.38 \pm 0.03$ | $6.21 \pm 0.05$ | $7.37 \pm 0.01$ | $6.21 \pm 0.01$ |
| K-Flow (ours) | $\mathbf{7.21 \pm 0.12}$ | $\mathbf{6.11 \pm 0.11}$ | $\mathbf{7.26 \pm 0.06}$ | $\mathbf{6.12 \pm 0.07}$ | $\mathbf{7.23 \pm 0.01}$ | $\mathbf{6.07 \pm 0.01}$ |

**Dataset and evaluation metrics.** We evaluate our method using the crystallization dataset COD-Cluster17 (Liu et al., 2024b), a curated subset of the Crystallography Open Database (COD)(Grazulis et al., 2009) containing 133K crystals. We consider three versions of COD-Cluster17 with 5K, 10K, and the full dataset. To assess the quality of the generated molecular assemblies, we employ *Packing Matching (PM)*(Chisholm & Motherwell, 2005), which quantifies how well the generated structures align with reference crystals in terms of spatial arrangement and packing density. Following (Liu et al., 2024b), we compute PM at both the atomic level (PM-atom) and the mass-center level (PM-center) (Chisholm & Motherwell, 2005).

**Baselines.** We evaluate our approach against GNN-MD (Liu et al., 2024b), variations of CrystalSDE and CrystalFlow (Liu et al., 2024b), and the state-of-the-art AssembleFlow (Guo et al., 2025). CrystalSDE-VE/VP model diffusion via stochastic differential equations, while CrystalFlow-VE/VP use flow matching, with VP focusing on variance preservation. CrystalFlow-LERP employs linear interpolation for efficiency. AssembleFlow (Guo et al., 2025) enhances rigidity modeling using an inertial frame.

**Main results.** The main results in Table S9 show that K-Flow outperforms all baselines across three datasets. Building on AssembleFlow's rigidity modeling, K-Flow decomposes molecular pairwise distances via spectral methods and projects geometric information from $\mathbb{R}^3$ and $SO^3$ accordingly. This approach achieves consistently superior packing matching performance.