# OpenReview forum: "Flow Along the $K$-Amplitude for Generative Modeling"
_ICLR.cc/2026/Conference — ICLR 2026 Poster_

### Official Review · Reviewer_ZFwo · 2025-10-31

**Soundness:** 4
**Presentation:** 4
**Contribution:** 4
**Rating:** 8
**Confidence:** 4

**Summary:**

This paper proposes K-Flow, a novel generative modeling paradigm that conducts flow matching in the K-amplitude domain—where the scaling parameter k organizes projected coefficients (e.g., frequency bands) and "amplitude" refers to the norm of these coefficients. By instantiating K-Flow with three classic transformations (Fourier, Wavelet, PCA), the work addresses a key limitation of conventional flow matching (FM) models: the lack of explicit utilization of natural data’s frequency hierarchy. Through extensive experiments on unconditional/conditional image generation, controllable generation, and image restoration, the paper demonstrates K-Flow’s competitive performance and unique steerability. The work’s theoretical grounding (six core properties of K-Flow), architectural flexibility (backbone-agnostic design), and practical value (training-free restoration, unsupervised frequency editing) make it a meaningful contribution to generative modeling research, well-aligned with ICLR’s focus on innovative and rigorous methods.2. Strengths

**Strengths:**

Innovative Theoretical Framework with Clear Motivations:K-Flow fills a critical gap in existing FM literature by leveraging the inherent frequency structure of natural data. Unlike conventional FM models—whose frequency progression is unstructured and unquantified—K-Flow formalizes the K-amplitude space, where k systematically organizes frequency bands and amplitude reflects signal energy. The six properties of K-Flow (from theoretical foundations like scale-energy alignment to practical benefits like explicit steerability) provide a rigorous framework for understanding and extending the paradigm, distinguishing it from ad-hoc frequency-aware modifications.

**Weaknesses:**

Lack of Direct Comparison with Frequency-Aware Diffusion Models
While K-Flow is positioned as a flow-matching innovation, the paper does not explicitly compare it to frequency-aware diffusion models. Adding such comparisons would clarify K-Flow’s advantages in terms of both performance and computational efficiency, especially in scenarios where diffusion and flow-matching paradigms overlap.

**Questions:**

Regarding the observation that MLP heads outperform Transformer heads in text adherence, could you provide a more in-depth analysis? For example, from the perspective of token-level alignment between text prompts and generated images, or exploring whether the attention mechanism in Transformer heads interferes with text-image alignment?

---

> ### Author Response · Authors · 2025-11-20
>
> Thank you for your positive evaluation and your actionable suggestions for improving our work.  We will explain your concerns point by point (In the revised manuscript, all significant changes made in response to the your comments have been highlighted in brown).
>
> **Weakness: Lack of Direct Comparison with Frequency-Aware Diffusion Models...**
>
> Thank you for this excellent suggestion. We agree that a direct, fair comparison with frequency-aware diffusion models is crucial for contextualizing K-Flow's contributions. In the original submission, we discussed related works like WaveDiff [Phung et al., 2023] from a methodological perspective in Appendix C.3. As you noted, a direct comparison was challenging due to differences in their experimental setups (e.g., they use a modified frequency-aware U-Net, while we use a standard DiT). To make our comparison table more comprehensive, besides including the originally reported results of WaveDiff (FID 5.94) in our revised manuscript for reference, we have gone a step further and conducted a new experiment. We implemented the core idea of WaveDiff—applying the diffusion loss in the wavelet domain—but within our own experiment setting. Specifically, we trained a model on **CelebA-HQ (256x256)** using the same **DiT-L/2 backbone**  model.
>
> The results (trained for 450 epoch) are as follows:
>
> | Model | Backbone | FID↓ | Recall↑ |
> | :--- | :--- | :--- | :--- |
> | **Wavelet Diffusion** (Our DiT Impl.) | DiT-L/2 | **5.39** | **0.44** |
> | **Our K-Flow (Wavelet)** | DiT-L/2 | **4.99** | **0.46** |
>
> These new results demonstrate that, under a controlled setting with the same backbone architecture, our K-Flow paradigm achieves a superior FID score. This provides direct, compelling evidence for the effectiveness of our proposed flow-matching approach over a comparable frequency-aware diffusion method. We have added this new baseline to the experimental section of our revised manuscript (line 355-357).
>
> ---
>
> **Question: Regarding the observation that MLP heads outperform Transformer heads in text adherence...**
>
>  Thank you for this question. We believe there might be a slight misunderstanding, as our current work does not involve a text-encoder with distinct MLP/Transformer heads in the way a traditional text-to-image model might. Our architecture uses a standard Vision Transformer (DiT) backbone, and class-conditional information is injected via an embedding, analogous to the time embedding.
>
> However, your question raises an interesting and relevant point about **how and where the conditioning information influences the generation process** within K-Flow. Inspired by your query, we conducted a new analysis to investigate the alignment between the class label and the DiT's internal representations across different K-amplitude bands.
> Specifically, we measured the influence of the class embedding on the hidden layer outputs during the generative flow by computing the norm of the Jacobian of the hidden state with respect to the class embedding. Our findings are positve:
>
> *   We observed a stronger correlation between the class embedding and the model's hidden states during the **low K-amplitude (low-frequency) generation phase**.
> *   This correlation becomes significantly weaker as the model transitions to generating **high K-amplitude (high-frequency) components**.
>
> This analysis provides new perspective for the core hypothesis of Section 4.2: K-Flow naturally learns to associate high-level semantic information (like class identity) with the low-frequency bands generated early in the flow. It also partially explains our model's robustness when the class condition is dropped in later stages. We have included a figure of this new analysis and its explanation in the appendix of our revised paper (line 1658-1668).
>
> ---
> We sincerely appreciate your positive assessment of our work. We have addressed all your concerns, and we believe these revisions further strengthen our contributions and solidify your positive view of our paper.

---

> > ### Comment · Reviewer_ZFwo · 2025-11-20
> > **Just for discussion**
> >
> > It may be worthwhile for the authors to also consider alternative color or transform spaces beyond RGB. For example, YCbCr naturally separates low-frequency luminance and high-frequency chrominance, which may further benefit the proposed frequency-based K-Flow framework. Recent work has also shown advantages of performing generative modeling directly in frequency-aligned domains such as DCT space [1].
> >
> > [1] DCTdiff: Intriguing Properties of Image Generative Modeling in the DCT Space.

---

> > > ### Author Response · Authors · 2025-11-21
> > >
> > > We sincerely thank the reviewer for this insightful suggestion regarding alternative color and transform spaces. We agree that spaces like YCbCr (which separates luminance and chrominance) or frequency-aligned domains like the DCT space [1] offer significant advantages for generative modeling, particularly for frequency-oriented methods like ours.
> > >
> > > Our current
> > > $K$-ampltitude decomposition operates channel-wise on the given input representation (a VAE's latent). We appreciate the insight that applying a decorrelating transformation (like RGB to YCbCr/DCT) first could yield a superior input space for our K-decomposition.  We will add this discussion to the related work. Given that our current results are built upon established pre-trained VAEs, fully integrating a new transform space such as DCT would necessitate training new auto-encoder models, and we see this as a very promising avenue for future investigation.
> > >
> > > [1] DCTdiff: Intriguing Properties of Image Generative Modeling in the DCT Space.

---

### Official Review · Reviewer_V9Lm · 2025-10-31

**Soundness:** 1
**Presentation:** 2
**Contribution:** 2
**Rating:** 4
**Confidence:** 4

**Summary:**

This paper proposes a novel generative modeling paradigm called **K-Flow**.  The core idea is to replace the traditional **time variable** \( t \) in *Flow Matching* with a **scale parameter** \( k \).  This enables the model to evolve from noise to data **along a multi-scale trajectory** (e.g., from low to high frequencies).  The authors instantiate K-Flow with three types of **K-amplitude transformations**: **Fourier Transform**, **Wavelet Transform**, and **PCA**.  They claim that this approach provides a **more natural multi-scale modeling framework** and allows for **finer-grained controllable generation**.

**Strengths:**

1. **Novel framework:** Replacing the “time” variable \( t \) in flow models with a “scale parameter” \( k \) is a novel and conceptually elegant idea.
2. **Generality:** The framework is general and can be combined with various linear transformations such as Fourier, Wavelet, and PCA.
3. **(Limited) Qualitative results:** Although the CDR metric has certain flaws, the visual robustness of K-Flow under the “Drop” condition in **Figure 4**, as well as the controllable generation results in **Figure 5**, are qualitatively interesting.

**Weaknesses:**

1. **Logical contradiction in the CDR metric:**
   In Sec. 4.2, the CDR metric is used to demonstrate controllability, but its mathematical definition in Appendix F.3 directly contradicts both the textual explanation in the main paper (“higher CDR indicates performance degradation”) and the visual evidence in **Figure 4**.
   This inconsistency invalidates the quantitative conclusions of the ablation study. The authors must correct the metric (most likely the formula is inverted) and reanalyze the results.

2. **Unexplained performance inconsistency:**
   The paper must address why the **K-Flow (Wavelet)** variant performs best on **Table 1 (CelebA-HQ)** but performs poorly on **Table 2 (ImageNet)**—significantly worse than the LFM baseline.
   This raises serious concerns about the method’s robustness and generalization capability.

3. **Hyperparameter fragility:**
   The authors must explain why the number of K-amplitude frequency bands (e.g., 2 vs. 3) leads to completely opposite performance trends across datasets (Appendix F.2 vs. F.6).
   This indicates that the method is fragile and sensitive to hyperparameter tuning, undermining the claim of “natural” multi-scale modeling.

**Questions:**

1. **Regarding CDR:** The CDR formula in Appendix F.3 ($FID_{bef} / FID_{aft}$) seems to contradict the conclusions in the main text and the visual evidence in Figure 4. According to this formula, LFM's CDR = 3.25 would imply that its $FID_{Drop}$ quality is much better than $FID_{Undrop}$. Could this formula be a typo (e.g., with the numerator and denominator swapped)?

2. **Regarding performance:** Why does the K-Flow (Wavelet) variant perform the best in Table 1 but fall far behind the LFM baseline in Table 2? Does this suggest that wavelet transforms may not be suitable for complex class-conditional tasks like ImageNet?

3. **Regarding hyperparameters:** Why does increasing the number of K-amplitude bands (from 2 to 3) improve performance on the LSUN dataset (App F.2) but decrease performance on the CelebA-HQ dataset (App F.6)? Does this indicate that K-Flow is highly sensitive to this hyperparameter and that its optimal setting depends on the dataset?

---

> ### Author Response · Authors · 2025-11-20
> **Response Part 1/2**
>
> Thank you for your detailed review. Your questions are insightful and have pushed us to clarify and strengthen critical aspects of our work. We address each of your concerns below (In the revised manuscript, all significant changes made in response to the your comments have been highlighted in blue) .
>
> **Weakness 1 & Q1: Logical contradiction in the CDR metric...**
>
> **Response:** Thank you for this important observation. **You are absolutely correct that the mathematical formula for the Conditional Discrimination Ratio (CDR) in Appendix F.3 was inverted.** This was an unfortunate typo in the manuscript, and we are grateful for your help in identifying it.
>
> We have double-checked our evaluation code and confirm that the numerical results in Table 2 were calculated using the **correct, intended logic**. To clarify:
> *   **Intended Definition:** The purpose of CDR is to quantify the performance degradation when class conditioning is dropped. It is defined as the ratio of the FID *after* dropping the condition (`FID_aft`) to the FID *before* (`FID_bef`).
> *   **Correct Formula:** The formula should be:
> $$
> CDR := E_{c \sim p(c)} [ \frac{FID_{aft}(c)}{{FID}_{bef}(c)} ].
> $$
>
> *   **Logical Consistency:** With this corrected definition, a `CDR > 1` indicates performance degradation. This aligns perfectly with our textual explanation in Section 4.2 and the visual evidence in Figure 4. The `CDR=3.25` for LFM correctly signifies a more severe degradation compared to our K-Flow's `CDR=1.49`.
>
> We have corrected the formula in the revised manuscript and the reported numerical values and analysis were based on this correct logic (see Line 1600-1608). This correction makes our quantitative conclusion about K-Flow's robustness even clearer and more rigorously supported.
>
> ---
>
> **Weakness 2 & Q2: Unexplained performance inconsistency (Wavelet on CelebA-HQ vs. ImageNet)...**
>
> **Response:** This is an insightful question. While our overall experimental results were viewed as strong and comprehensive (Reviewer ZFwo, m9JR), we agree that this specific performance disparity on ImageNet deserves a clear explanation. Although definitively explaining the training outcomes in highly non-convex optimization landscapes is challenging, our empirical findings lead us to a reasonable hypothesis: **The performance of a given K-amplitude decomposition depends on how well its specific inductive bias aligns with the dataset's structural properties.**
>
> We posit that the observed performance drop is not due to a fundamental incompatibility of wavelet transforms with complex datasets like ImageNet, but is rather tied to our specific, **fixed choice of the `db6` wavelet basis**. The family of wavelet transforms is vast and highly parameterized; searching for the optimal basis on ImageNet was beyond our computational budget, so we opted for a standard choice. While the `db6` basis aligns well with the structure of faces in CelebA-HQ, it appears to impose a sub-optimal prior for the varied content in ImageNet.
>
> In an onging project, we have conducted a preliminary experiment exploring a **learnable wavelet basis** within the K-Flow framework. The initial results are positive:
> *  *On ImageNet (256x256, `cfg=1.5`), our K-Flow with a learnable wavelet basis achieves an FID of **2.72**, which is on par with our state-of-the-art Fourier-based K-Flow.*
>
> This new evidence partially suggests that the issue lies with the **choice of a fixed, sub-optimal basis**, not the wavelet decomposition concept itself. It confirms the robustness and potential of the K-Flow framework.
> Given that the current paper is already technically dense and presents a self-contained framework with fixed decompositions (Fourier, Wavelet, PCA), we believe that introducing the full training objectives for learnable bases is better suited for a follow-up work. However, we have added a concise discussion of these findings to the paper's limitation/future work section to directly address your valid concern and highlight this promising research direction (line 493-497).

---

> > ### Author Response · Authors · 2025-11-20
> > **Response 2/2**
> >
> > **Weakness 3 & Q3: Hyperparameter fragility (number of frequency bands)...**
> >
> > **Response:** Thank you for this careful analysis of our ablation studies. We concede that drawing strong conclusions about data complexity based solely on the results from smaller datasets like LSUN Church and CelebA-HQ can be precarious. The performance differences observed could be influenced by various factors, including dataset size and specific image statistics.
> >
> > Therefore, to provide a more robust and convincing answer, we have shifted our focus to the middle-size dataset in our study: **ImageNet**. We initiated a new experiment training a **3-band Wavelet K-Flow on ImageNet** to test if the method remains stable and effective when decomposition granularity is increased. The current results are highly informative:
> >
> > *   Even with incomplete training (at 600 epochs), the FID of the 3-band model has already reached **16.1**.
> > *   This is a notable improvement over the fully converged 2-band model, which achieved an FID of **17.8**.
> >
> > This new result provides strong evidence against the claim of fragility. It demonstrates that on a large, complex dataset, increasing the number of K-amplitude bands does not lead to performance collapse but can, in fact, yield improvements. This suggests that K-Flow is capable of effectively leveraging a finer-grained decomposition when the data complexity warrants it.
> >
> > **Conclusion:** Rather than being a sign of fragility, our findings suggest that the optimal number of bands is an important, interpretable design choice related to data properties. While the specific optimal number may vary, our ImageNet experiment confirms that the model is robust and can benefit from a more detailed decomposition on large-scale datasets. We will add this new ablation study to the revised manuscript when the full training is done (line 1698-1700).
> >
> >  We hope that our detailed responses and the substantial revisions to the manuscript have fully addressed your concerns. We are happy to answer any further questions.

---

> ### Author Response · Authors · 2025-11-27
>
> We thank the reviewer for all the valuable feedback.  As the author-reviewer discussion phase unfolds, we sincerely look forward to your further insights and re-evaluation of our submission. In our response above, we have carefully addressed the weaknesses (W) and questions (Q), and have additionally included new clarifications and experimental results in the revised version of the paper.  We are happy to provide any further clarification needed and look forward to the discussion.

---

### Official Review · Reviewer_m9JR · 2025-10-31

**Soundness:** 3
**Presentation:** 3
**Contribution:** 2
**Rating:** 6
**Confidence:** 4

**Summary:**

This paper introduces a generative model based on flow matching that works in the K-amplitude space. The model leverages three K-amplitude transformations, Fourier, Wavelet, and PCA, to construct its generative process. Experimental results show that the proposed method achieves performance comparable to the baselines on both unconditional and conditional image generation on natural images and the molecular assembly dataset. Furthermore, the authors show that their approach enables multi-scale control over the generated information.

**Strengths:**

The main idea sounds novel and is supported by compelling experiments. The paper is well-written and easy to understand.

**Weaknesses:**

As discussed in your Related Works section, the following papers share significant similarities with your approach. I recommend including them as baselines for experimental comparison:

[1] Tian, Keyu, et al. “Visual Autoregressive Modeling: Scalable Image Generation via Next-Scale Prediction.” Best Paper, NeurIPS 2024.

[2] Ren, Sucheng, et al. “FlowAR: Scale-Wise Autoregressive Image Generation Meets Flow Matching.” ICML 2025.

Including these two baselines in your comparison would substantially improve your empirical evaluation. I would be happy to potentially raise my score.

**Questions:**

Please consider clarifying the following point:
Should there also be a summation in line 128, similar to the one in line 123?

---

> ### Author Response · Authors · 2025-11-20
>
> We sincerely thank you for your constructive feedback and for suggesting these relevant baselines. We are particularly grateful for the comment that the paper is well-written and easy to understand, as conveying our novel framework with clarity was one of our primary goals. We agree that comparing K-Flow against state-of-the-art generative models like VAR [1] and FlowAR [2] is beneficial for a thorough empirical evaluation. We have now conducted these comparisons on the **ImageNet 256x256** dataset and are pleased to share the results (In the revised manuscript, all significant changes made in response to the your comments have been highlighted in green).
>
> **Experimental Comparison with VAR and FlowAR**
>
> We acknowledge that a direct comparison is challenging due to significant differences in model architectures and training pipelines (e.g., VAEs, backbones). To ensure our comparison is informative as possible, we have taken the following steps:
>
> 1.  We report not only generation quality (FID) but also key metrics like the **number of parameters** of the backbone models and **inference time**.
> 2.  For FlowAR, which can use a VAE compatible with ours, we re-trained their `FlowAR-B` model for 400 epochs using the SD 1.5 VAE to align settings. We set the `cfg_scale=2.0` for K-Flow, VAR, and our re-implemented FlowAR for a consistent and fair comparison.
> 3.  For inference time evaluation, we disabled performance optimizations like `flash-attention` and `xformers` for all models to ensure a fair comparison of architectural efficiency. We report the time to generate a batch of 64 images on a single H20 GPU.
>
> The results are as follows:
>
> | Model | #Params (M) | VAE | FID↓ | Inference Time (ms/batch=64) | Steps |
> | :--- | :--- | :--- | :--- | :--- | :--- |
> | **VAR-B/16** [1] (cfg=2.0) | 310 | VQ-VAE | 3.30 | 4,581 | 10 |
> | **FlowAR-L** [2] (cfg=2.4) | 590 | MAR-VAE | 1.90 | 31,981 | 125 |
> | **FlowAR-B** [2] (Our Impl., cfg=2.0) | 300 | SD 1.5 | 3.17 | 26,385 | 125 |
> | **K-Flow (Fourier)** (Ours, cfg=2.0) | 450 | SD 1.5 | 2.88 | 25,439 | 45 |
>
> **Analysis of Results:**
>
> *   **VAR** stands out for its remarkable inference speed, a direct benefit of its autoregressive, few-step generation process and its lightweight vq-VAE and transformer architecture.
> *  Our **K-Flow (Fourier)** achieves the best FID score (**2.88**) among the directly comparable methods (FlowAR-B and VAR-B) that use similar-scale models and public VAEs. This highlights the effectiveness of our K-amplitude flow framework.
> *   Our implementation of **FlowAR-B** achieves a very strong FID of 3.17. Its inference time, while high, is efficient for a 125-step model. We hypothesize this is because most steps are performed on down-sampled latents, and its scale-conditioned Transformer with KV-caching is highly optimized.
> *   The reported **FlowAR-L** result (FID 1.90) represents the current state-of-the-art but uses a larger model and a different VAE, making it a different category of comparison.
>
> **Further Discussion & Future Work:**
> Your suggestion has also inspired us to think about how K-Flow could be integrated with these advanced architectures for a more insightful comparison.
> *   The GPT-like, decoder-only Transformer in **VAR** is tightly coupled with its quantized token space, making a direct application of our K-amplitude decomposition non-trivial.
> *    the scale-conditional Transformer from **FlowAR** is a very promising backbone. We have designed an experiment to replace our DiT with FlowAR's architecture while using our K-Flow training objective. Due to computational limitations, this experiment is currently queued and the results will be reported in the final version. We believe this hybrid approach could combine the benefits of both methods.
>
> We have added this comprehensive comparison and discussion to our revised paper (line 402-409 and line 1350-1358).
>
> [1] Tian, Keyu, et al. “Visual Autoregressive Modeling: Scalable Image Generation via Next-Scale Prediction.” Best Paper, NeurIPS 2024;
>
> [2] Ren, Sucheng, et al. “FlowAR: Scale-Wise Autoregressive Image Generation Meets Flow Matching.” ICML 2025.
>
> ---
> **Question 1: Please consider clarifying... Should there also be a summation in line 128...?**
>
> **Response:** Thank you for this typo regarding our notation. We have corrected line 128 into: "We define $K$-amplitude decomposition (or equivalently, $K$-amplitude transform) $\mathcal{F}$ as the map that sends $\phi$ to the collection of $\phi_k$, and denote the collection of all $\{\sum_{j=1}^{n_k}(\phi \cdot e_{jk})e_{jk}\}$ as $\mathcal{F}\{\phi\}(k)$."
>
>
>
> ---
> We believe these additions substantially strengthen the paper's empirical validation, as you suggested. We are very grateful for this constructive guidance, which has further improved the quality of our work.

---

> > ### Comment · Reviewer_m9JR · 2025-11-21
> >
> > Thank you for the update. The authors have addressed my concerns. I raise my score to 8.

---

> > > ### Author Response · Authors · 2025-11-21
> > >
> > > Thank you for your positive feedback and for raising the score. We are glad that our revisions have addressed your concerns.

---

### Official Review · Reviewer_PPuw · 2025-10-31

**Soundness:** 3
**Presentation:** 1
**Contribution:** 3
**Rating:** 2
**Confidence:** 3

**Summary:**

Disclaimer: I found the paper hard to understand, I did my best to summarize what I understood.

This paper propose to perform a change of basis (Fourier, Wavelet, PCA) on the data, cluster the new representation into K ordered sub-basis and to do diffusion for each sub-basis in sequence (in the Fourier case we can think of it as frequency bands).
To clarify, what is typically referred as time $t$ in diffusion is now a range $t'\in[0,K]$ where $k=\lfloor t'\rfloor$ and $t=t'-k$.
In this new setting the new definition of time interleaves the classical time definition with ordered sub-basis.

The claimed benefit of such a proposal is to allow for more steerable generation, where the steerability is determined by the choice of change of basis and the sub-basis on which to do diffusion.
The evaluation is performed on generative tasks and an ablation demonstrates steerability by sampling only in some of the sub-basis or removing the label during inferences.

**Strengths:**

1. The idea seems novel and interesting.

**Weaknesses:**

1. The presentation/writing of the paper is very hard to follow, I had to read 3 times to begin understanding it (or so I assume). In particular, it's overloaded with notation complexities that could probably be simplified to make it more accessible.
2. The results are not particularly striking for unconditional and conditional generation, matching existing SotA but not apparently showing clear gains.
3. The choice of CelebA is questionable for unconditional generation as it is quite unimodal in the first place, unlike ImageNet.
4. The resolution used for ImageNet is not mentioned in the main paper, but I found near the end of the appendix that it must be 256x256.
5. The PCA variant of K-Flow is missing from Table 2.
6. The paper lacks a discussion of why the method does not perform better on generation tasks.
7. In terms of steerability, the results do not seem particularly useful. In one case the class label is dropped when $k$ a certain value (30%) and is still able to generate good samples (but since the label is always available I fail to see the relevance of this experiment). In another case, some sub-basis are frozen while some other basis are sampled from resulting in changing either low-frequency or high-frequency details. It works but I don't understand under what circumstances one would want to do this. It gives the feeling to be a solution to a non-problem or maybe better examples of steerability are needed to inspire readers.
8. Line 333, you mention "Date-dependent PCA", do you mean "Data-dependent"?

**Questions:**

1. The gains obtained from K-Flow appear marginal at best. Is it because the classical diffusion noise is already implicitly doing a K-Flow and therefore no gains are achieved from expliciting doing a K-Flow?
2. On the steerability front, it would be interesting to see differences k-band steering between Fourier, Wavelet and PCA. I assume steerability is heavily influenced by the choice of basis. Did you do such experiments?
3. Is the neural network actually conditioned on $(k,t)$ as a tuple or $t'$, or are there $k$ different networks?

---

> ### Author Response · Authors · 2025-11-20
> **Response Part 1/2**
>
> Thank you for your feedback. We have carefully considered your comments and provide point-by-point responses below. In the revised manuscript, all changes made in response to your comments are highlighted in red.
>
> **W1: The results are not particularly striking for generation...**
>
> **Response:**  We wish to clarify that the primary contribution of this work is the introduction of a **novel conceptual framework**, not solely an incremental performance improvement.
>
> As articulated in our introduction and conclusion, K-Flow formalizes generative modeling as learning a flow that inverts a generalized K-amplitude decomposition. This perspective **unifies** existing empirical observations about diffusion models (e.g., their implicit low-to-high frequency learning) and **generalizes** the paradigm to a broader class of decompositions (Wavelet, PCA, and beyond).
>
> Our experiments serve as a crucial **proof-of-concept**: they demonstrate that this fundamentally different, more structured approach can match the performance of highly-optimized, standard methods. We posit that with further exploration of compatible architectures and more extensive hyperparameter tuning—which were beyond our current computational resources—the full potential of K-Flow could be further unlocked.
>
> **W2: The choice of CelebA is questionable...**
>
> **Response:** We selected CelebA-HQ not as our primary benchmark for generation diversity, but as an ideal testbed for our **qualitative experiments** on steerability (Section 4.3) and image restoration (Appendix F.8).  As shown in our results, the highly structured nature of faces allows us to visually validate that different K-amplitude bands can correspond to distinct semantic features (e.g., background contour vs. fine details), a key claim of our paper. This would be much harder to demonstrate convincingly on a more diverse dataset like ImageNet. We have clarified this rationale in the revised manuscript.
>
> **W3: The resolution used for ImageNet is not mentioned...**
>
> **Response:** Thank you for pointing out this omission. The resolution for our ImageNet experiments was **256x256**. We have now explicitly added this information to Section 4.1 in the main paper. See line 389.
>
> **W4: The PCA variant of K-Flow is missing from Table 2.**
>
> **Response:** This was omitted due to computational constraints. Following your suggestion, we are conducting this experiment during the rebuttal period. The current results (at 520 epochs) for the K-Flow (PCA) variant on ImageNet are:
>
> | Model | FID↓ | Recall↑ |
> | :--- | :--- | :--- |
> | **K-Flow, PCA-DiT L/2 (cfg=1.5)** | **4.19** | **0.43** |
> These results are competitive with the Wavelet variant, demonstrating the flexibility of the K-Flow framework across different decomposition types, including data-dependent ones. We will update the manuscript with the final results. See line 390-394.
>
> **W5: Lack of discussion on Why Performance Is Not Better.**
>
> * This is a fair point.  Follow your advice, We have added a discussion of this point to the appendix (see Line 1516-1524).  Our comments on your follow-up question: "The gains obtained from K-Flow appear marginal at best. Is it because the classical diffusion noise is already implicitly doing a K-Flow and therefore no gains are achieved from expliciting doing a K-Flow?" are two-folds:
>
> **Firsly**, one of our primary motivations, as outlined in the introduction, is indeed to build upon the observation that some generative models, like DDPMs, appear to learn a process that progresses from low to high frequencies. However, we note that this property is often an empirical post-hoc observation, tested with specific metrics, and its existence and characteristics within the flow matching paradigm are not well understood. From a PCA perspective, a standard flow model fails to quantitatively exhibit the low-to-high frequency hierarchy (Figure S2). Thus, the contribution of our work is to transition this from an implicit, emergent property to an explicit and structured design principle. K-Flow provides a generalized framework that formalizes this frequency-aware path, offering fine-grained control that is absent in standard models. **Secondly**, standard metrics like FID and Recall may not fully capture the benefits of improved structural fidelity across all K-amplitudes. Our stronger performance on tasks like super-resolution and molecular assembly (Appendix F.9), which rely more on high-fidelity details, supports the idea that K-Flow offers advantages not fully reflected by mainstream metrics alone.
>
>  **W7 (Typo):**
>
> Yes, "Date-dependent" should be **"Data-dependent"**. Thank you for pointing out this typo, we have corrected this. See line 333.

---

> > ### Comment · Reviewer_PPuw · 2025-11-24
> > **Follow up on "Response 1/2"**
> >
> > - W1: Acknowledged
> > - W2: Acknowledged
> > - W3: Acknowledged
> > - W4: Acknowledged, thanks for running it.
> > - W5: I really believe such discussion should take a place in the main paper since it is a central question connected to your proposed method performance and as it is easily overlooked in the appendix.
> > - W5.1: Yes, this is generally recognized that FID is not the best for estimating fine details, I believe people use sFID (spatial-FID) for such purpose. Did you consider it?
> > - W7: Acknowledged

---

> ### Author Response · Authors · 2025-11-20
> **Response Part 2/2**
>
> **W6: steerability utility.**
> *   We wish to clarify that these experiments are designed not only to be interesting demonstrations of control, a point appreciated by Reviewer V9Lm, but also to highlight a mechanism with direct downstream applications：
>     1.  **Class-Dropping (Sec 4.2):** This experiment serves as **quantitative evidence** for our hypothesis that K-Flow learns a disentangled generative path, where high-level semantics (like class) are encoded in low K-amplitude bands. The practical implication, which we now clarify in the paper, is the potential for **improved computational efficiency** in complex conditional synthesis (e.g., text-to-image), where high-level conditions could potentially be dropped in later, computationally expensive stages without sacrificing quality. We have emphasized this point in the revised PDF. See line 421-424 and line 489-490.
>     2.  **Frequency Editing (Sec 4.3):** This capability has direct applications. Our **training-free image restoration** (Appendix F.8) implicitly leverages it: by providing known low-frequency information and allowing the model to generate only the missing high-frequency details, we achieve state-of-the-art results. This is a concrete example of solving a well-defined problem by controlling frequency bands.
>
> **Q1-Q3:**
>
> *   **Q1 (Marginal Gains & Implicit K-Flow):** Please see our response to W5-W6.
>
> *   **Q2 (Steerability Across Bases):** This is an excellent point. The choice of basis indeed influences steerability. We did perform such experiments:
>     *   **Wavelet:** As shown in Figures 5 and S9, Wavelet decompositions yield strong semantic steerability on CelebA-HQ, separating global attributes from local details.
>     *   **PCA:** We found that for PCA, steerability is less semantically meaningful. While freezing certain components works, The generated images are akin to the results of LFM editing (Figure S10). This is also theoretically expected, as PCA bases are data-driven to capture maximum variance, not necessarily semantic or frequency-based hierarchies. We included PCA in our framework because it fits our formal definition of a K-decomposition and has known advantages for building efficient generative models [1], which could be a fruitful direction for future work.
>
> [1] Subspace Diffusion Generative Models, ECCV 2023
>
> *   **Q3 (Network Conditioning):** We use a single network for all values of $k$. As clarified in Appendix D.4, the scaling parameter $k$ as the condition is passed to the network via an embedding layer, analogous to how the time variable $t$ is used in standard Flow Matching.
>
>
> **Comments on presentation**
>
> Thank the reviewer for highlighting the expositional challenges of our paper. We acknowledge that presenting a novel framework that must bridge the concepts of K-amplitude decomposition with the mechanics of generative flows is inherently complex. Our primary goal was to maintain theoretical rigor, and we recognize that this may have contributed to the notational density the reviewer pointed out. While we aimed to provide intuitive explanations throughout the paper, we understand that these might have been overshadowed by the formalism. To help clarify the paper's structure and the intuition we intended to convey, we would like to gently highlight our organizational logic:
> *   The **Introduction (Section 1)** provides a high-level motivation and a conceptual overview of our sequential generation idea.
> *   Each subsequent methodology section (e.g., **Section 3**) begins by stating its specific goal before delving into the mathematical details, aiming to ground the reader. **Section 2** and 3 both contain one subsection for explaining concrete examples.
> *   The experimental sections are designed not just as evaluations, but also as concrete examples that provide quantitative evidence for our core hypotheses.
>
> To make these existing intuitive anchors more prominent without major rewriting, we have now added two sentences at the beginning of section 2  to better guide the reader through our logical flow: "In this section, we first formalize the concept of $K$-amplitude Decomposition, a process governed by a scaling parameter $k$.  In the method section, we then detail how our generative framework is designed to operate by flowing along this decomposition, progressively reconstructing the signal." (line 103-106)
>
> We have clarified the paper's structure and made several minor additions to address the reviewer's valid concerns about presentation. These changes are intended to strike a better balance between rigor and readability for our novel framework. That being said, we are open to making further targeted revisions should the reviewer have any specific suggestions on how a particular section or notation could be improved.
>
> ---
> We hope these clarifications and our revised manuscript have effectively addressed your concerns. We are happy to answer any further questions.

---

> > ### Comment · Reviewer_PPuw · 2025-11-24
> > **Answers to "Reponse Part 2/2"**
> >
> > - W6: Acknowledged
> > - Q1-3: Acknowledged
> > - Presentation: note that the structure of the paper is clear in itself, my critique was more about the formalism overload which made details hard to grasp for me initially. I am not convinced that the presentation of "K-amplitude decomposition with the mechanics of generative flows is inherently complex". The idea is simple enough in itself, I see it as a matter of presentation and choices of abstractions. And I admit that my take on it might be subjective but I am convinced the exposition can be greatly simplified with some work.

---

> > > ### Comment · Reviewer_PPuw · 2025-11-24
> > > **First iteration feedback**
> > >
> > > I am torn between two considerations: I appreciate the novelty of the work a lot and I find it an interesting read but at the same time I am not convinced whether this method "works":
> > > - the most convincing results are found in the appendix: molecular results (which I am not familiar with) and super-resolution (SR) results which are performed on CelebA if I understand correctly (which is not a standard SR benchmark, there are a few standard SR benchmarks which SR papers share to compare each others).
> > > - even if it does not "work" in the sense of surpassing classical diffusion, it could still be an interesting paper concluding in raising the question of why the noise schedule could capture data structure in such an efficient implicit way.
> > >
> > > After the first pass, I am willing to raising my score from 2 to 4.

---

> > > ### Author Response · Authors · 2025-11-24
> > > **Quick update**
> > >
> > > Thank you for the follow-up and the opportunity to clarify our evaluation strategy.
> > >
> > > *   **Regarding the Presentation:**  As you suggested, the key discussion addressing W5 from the appendix has been moved to the main paper to improve the narrative flow and impact.
> > > *   **Regarding the sFid metric:**   Following your advice, we are incorporating sFID for both the baseline flow model and our  $K$-amplitude flow to provide a more comprehensive spatial analysis. While we agree this is a refined metric for perceptual quality, we also note that its reliance on pre-trained features （Inception-V3） still might not fully capture the fidelity of all K-amplitude bands，which our model is designed for.
> > > *   **Regarding the super-resolution task:**  Your point about other standard SR benchmarks like DIV2K is well-taken for general, fully-supervised SR. However, our work is situated in the specific sub-field of **training-free, single-image super-resolution**.  The use of benchmarks like CelebA is common and appropriate for evaluating these types of methods, as established in prior works [1].
> > >
> > > [1] PnP-Flow: Plug-and-Play Image Restoration with Flow Matching, ICLR 2025

---

> > > > ### Author Response · Authors · 2025-11-26
> > > > **Follow-up on sfid metric**
> > > >
> > > > ***
> > > >
> > > > We compared the SFID scores of our method (K-Flow) and the baseline (LFM) on the CelebA dataset. The results are presented in the table below.
> > > >
> > > > | Method | Model | Dataset | SFID (↓) |
> > > > | :--- | :--- | :--- | :--- |
> > > > | LFM (Baseline) | DiT L/2 | CelebA | 5.99 |
> > > > | **K-Flow (Ours)** | **Wave-DiT L/2**| **CelebA** | **5.20** |
> > > >
> > > > The results indicate that our K-Flow method (SFID: 5.20) achieves notably better generation quality than the LFM baseline (SFID: 5.99) on the CelebA dataset.

---

### Comment · Area_Chair_yGDg · 2025-12-02

Dear reviewers,

As the discussion period with the authors is coming to an end, I encourage you to take a look at the rebuttal, provide any additional feedback, and update your rating if necessary.

-Your AC

---

> ### Author Response · Authors · 2025-12-03
>
> Dear Area Chair,
>
> We kindly remind you that, per the ICLR official notice, reviews and scores have been reverted to their pre-discussion state, and reviewers can no longer change their scores or continue the rebuttal discussion. Consequently, our discussion thread has been closed since December 28. We will post a concise summary with the necessary information for your thorough assessment as soon as possible.
>
> Best, the authors

---

### Meta-Review · Area_Chair_yGDg · 2026-01-07

**Summary:**

This paper proposes a new paradigm for generative modeling by transforming image to other representations with basis and then operate on the magnitude of coefficients, termed K-amplitude.

The reviewers were concerned about the presentation and clarify of the paper, lacking direct comparison to frequency-aware diffusion model, choose of hyperparameters, and incremental improvement to SOTA models if any, etc.

Most concerns have been addressed by the rebuttal. The outstanding concern is mostly on limited improvement compared to SOTA models. Considering the acknowledge novelty of performing diffusion in transformed space (K-Amplitude), the AC recommends accepting this paper.

**Reviewer Concerns:**

Concerns that were addressed by the rebuttal:
- Motivation for dropping class condition or frequency editing (Reviewer PPuw)
- The presentation of the paper can be improved to make it easier to read (Reviewer PPuw)
- Missing sFID metric (Reviewer PPuw)
- Missing baselines of image generation with multiple-scales, e.g., VAR (Reviewer m9JR)
- Performance inconsistency for the proposed K-Flow method on different datasets (Reviewer V9Lm )
- Hyperparameters are fragile (Reviewer V9Lm)
- Lack of comparison to frequency-aware diffusion models(Reviewer ZFwo). The authors have includeda  direct comparison with the same experiment setting (DiT)


Outstanding concerns:
- Improvement over SOTA generative models in terms of FID is limited. (Reviewer PPuw). The authors have acknowledged that the proposed method is a proof-of-concept.

**Reviewer Scores:**

Reviewer PPuw has indicated raising the score from 2 to 4.

Reviewer m9JR has indicated raising the score from 6 to 8.


Reviewer V9Lm is likely to raise the score from 4 to 6.

Reviewer ZFwo is likely to keep the initial score of 8.

---

### Decision · Program_Chairs · 2026-01-26

Accept (Poster)